# Distinct representation of cue-outcome association by D1 and D2 neurons in the ventral striatum's olfactory tubercle

**Nuné Martiros, Vikrant Kapoor, Spencer E Kim, Venkatesh N Murthy***

Department of Molecular & Cellular Biology and Center for Brain Science, Harvard University, Cambridge, United States

**Abstract** Positive and negative associations acquired through olfactory experience are thought to be especially strong and long-lasting. The conserved direct olfactory sensory input to the ventral striatal olfactory tubercle (OT) and its convergence with dense dopaminergic input to the OT could underlie this privileged form of associative memory, but how this process occurs is not well understood. We imaged the activity of the two canonical types of striatal neurons, expressing D1- or D2-type dopamine receptors, in the OT at cellular resolution while mice learned odor-outcome associations ranging from aversive to rewarding. D1 and D2 neurons both responded to rewarding and aversive odors. D1 neurons in the OT robustly and bidirectionally represented odor valence, responding similarly to odors predicting similar outcomes regardless of odor identity. This valence representation persisted even in the absence of a licking response to the odors and in the absence of the outcomes, indicating a true transformation of odor sensory information by D1 OT neurons. In contrast, D2 neuronal representation of the odor-outcome associations was weaker, contingent on a licking response by the mouse, and D2 neurons were more selective for odor identity than valence. Stimulus valence coding in the OT was modality-sensitive, with separate sets of D1 neurons responding to odors and sounds predicting the same outcomes, suggesting that integration of multimodal valence information happens downstream of the OT. Our results point to distinct representation of identity and valence of odor stimuli by D1 and D2 neurons in the OT.

***For correspondence:**
vnmurthy@fas.harvard.edu

**Competing interest:** The authors declare that no competing interests exist.

## Editor's evaluation

Martiros et al. monitored the activity of dopamine D1 and D2-expressing neurons within the ventral striatum olfactory tubercle using 2-photon microscopy, as mice learned to associate odors and tones with either positive or aversive outcomes. Authors find differential roles for these neurons in the learning of odor valence and tone outcome associations. Overall, this study was deemed as an interesting and important contribution to our understanding of the neural basis of cue encoding.

## Introduction

Assigning value to stimuli in the external environment and subsequently adjusting behavior on the basis of those learned values is a primary function of the nervous system. Understanding this process is especially important because certain associations result in maladaptive behaviors such as compulsions, binge eating, and drug addiction (*Everitt and Robbins, 2005*; *Keiflin and Janak, 2015*; *Wise and Koob, 2014*). Olfaction or chemosensation is thought to have preceded other sensory modalities in evolution as the first direct link of the nervous system to the external environment (*Ache and Young, 2005*; *Kaas, 2008*; *Schneider, 2013*) to allow for such stimulus value learning. Olfactory sensory pathways to the forebrain remained relatively conserved across phyla while visual, auditory,

and somatosensory information was routed in polysynaptic pathways through thalamus and cortex (*Purves and Fitzpatrick, 2001*; *Schneider, 2013*). Due to this, the olfactory system has uniquely direct access to limbic centers of the brain. In particular, the olfactory tubercle (OT), a ventral basal ganglia structure known to be involved in reward processing (*Gadziola et al., 2015*; *Hagamen et al., 1977*; *Ikemoto, 2003*; *Ikemoto, 2007*; *Wesson and Wilson, 2011*; *Zhang et al., 2017*), is a direct recipient of a stream of olfactory input from a distinct class of olfactory bulb neurons, the tufted cells (*Igarashi et al., 2012*). In addition to olfactory sensory input, the OT receives dense dopaminergic input from the ventral tegmental area (VTA), similar to other ventral striatal regions – suggesting that the OT is ideally suited to integrate these inputs to assign valence to olfactory stimuli based on experience. It has been described in humans that olfactory sensory cues may be more powerfully linked to emotional memories than other types of sensory cues (*Miles and Berntsen, 2011*; *Pointer and Bond, 1998*; *Reid et al., 2015*), and that odor associations can be important in psychological health and disease (*Daniels and Vermetten, 2016*; *Herz, 2016*), suggesting a possible specialized form of olfactory memory in limbic regions such as the OT.

The OT has been implicated in motivation as well as olfactory processing (*Hagamen et al., 1977*; *Ikemoto, 2007*; *Koob et al., 1978*; *Koob and Volkow, 2010*; *Wesson and Wilson, 2011*; *Wright and Wesson, 2021*). As an example, self-administration of cocaine into the OT was found to be even more reinforcing than administration into nucleus accumbens, a key region known to be involved in reward, motivation, and addiction (*Ikemoto, 2003*). Stimulation of OT neurons or dopaminergic terminals in the OT has been shown to be effective in inducing odor preference or approach (*Fitzgerald et al., 2014*; *Gadziola et al., 2020*; *Zhang et al., 2017*). Electrophysiological recordings in the OT have revealed that OT neurons differentiate between rewarded and unrewarded odors in a go/no-go task, and quickly track changing odor-outcome contingencies (*Gadziola et al., 2015*; *Millman and Murthy, 2020*). These effects may extend to humans with elevated OT activity and development of place preference in response to attractive odorants (*Midroit et al., 2021*). Moreover, the OT appears to be the olfactory processing site most strongly involved in tracking odor-outcome associations, as direct comparison to recordings in posterior piriform cortex revealed weaker representation of odor-reward contingency (*Gadziola et al., 2020*; *Millman and Murthy, 2020*). The much higher density of dopaminergic input to the OT as compared to piriform cortex is likely relevant in differentiating the functions of these two parallel olfactory processing regions.

This evidence establishes OT as a key region likely involved in learning odor-outcome associations and assigning emotional tags to odors. However, many questions remain unanswered regarding the nature of the information encoded by OT neurons and the role of the different neuronal types within the OT in this function. OT is a part of the ventral striatum and consists of spiny projection neurons (SPNs). Striatal SPNs are roughly divided into D1-type and D2-type dopamine receptor expressing SPNs, and these two groups of SPNs also have different output projection patterns (*Bolam et al., 2000*; *Gerfen and Surmeier, 2011*). This differentiation of OT projection neurons into D1 and D2 type is especially relevant when considering the role of dopaminergic input in shaping odor valence representation. D1- and D2-type dopamine receptors are thought to differ in terms of their response to dopamine, in particular the plasticity rules regulated by dopamine (*Gerfen and Surmeier, 2011*; *Lovinger, 2010*; *Nicola et al., 2000*). To address the role of these neuronal subtypes in the OT, we conducted the first (to our knowledge) two-photon imaging of specific neuronal types in the OT in behaving mice. We then used experimental manipulations to address questions about the role of the OT and these neuronal types in the stimulus identity-valence-response transformation function.

## Results

To investigate the role of D1 and D2 receptor expressing neurons in the OT in odor valence learning, we imaged the activity of these neuronal populations in the OT using multiphoton microscopy, while mice learned to associate odors with aversive or rewarding outcomes. Adenoassociated virus that allows conditional expression of the calcium indicator GCaMP7s in the presence of Cre recombinase was injected into the right OT of Drd1-Cre and Adora2A-Cre mice and a 1 mm cannula was implanted targeting the OT (*Figure 1A*, *Figure 1—figure supplement 1*). After recovery, a Gradient-Index (GRIN) lens was placed in the cannula, mice were water restricted, habituated to head fixation, and trained on an odor-outcome conditioning task. Five monomolecular odors were coupled with graded outcomes ranging from aversive to rewarding (strong airpuff to nose, weak airpuff to

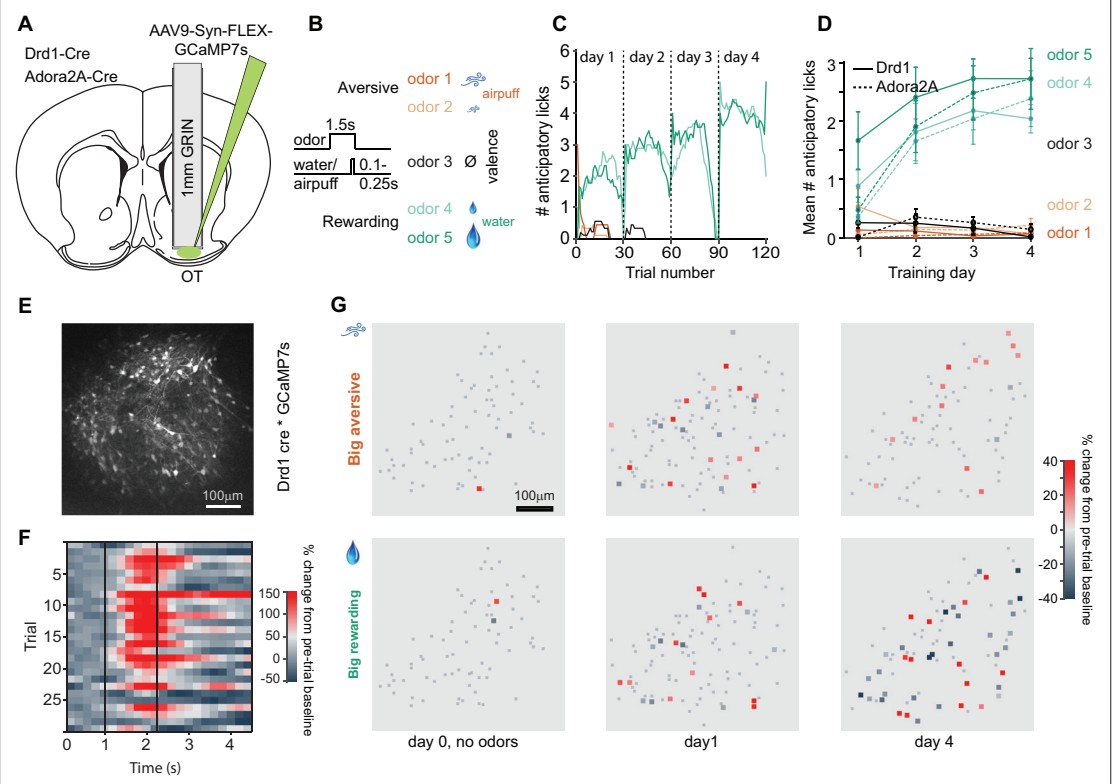

**Figure 1.** Two-photon calcium imaging of D1- and D2-type neurons in the olfactory tubercle (OT) during odor-outcome association learning.
(**A**) Cannula and Gradient-Index (GRIN) lens implantation targeting the OT in Drd1-Cre and Adora2A-Cre mice with AAV9-Syn-FLEX-GCaMP7s virus injection in the OT. (**B**) Odor-outcome task structure, odors 1–2 are paired with aversive airpuffs and odors 4–5 are paired with rewarding water drops in headfixed water restricted mice. Odor-outcome assignments are counterbalanced across mice. (**C**) Number of anticipatory licks in an example mouse in a 1 s period after odor onset and prior to water or airpuff delivery. Each training day has 30 trials of each of the five odors. (**D**) Mean number of anticipatory licks across 4 days of training in implanted Drd1-Cre mice (n=6, solid) and Adora2A-Cre mice (n=6, dashed). (**E**) Field of view of GCaMP7s expressing neurons in a Drd1-Cre mouse. (**F**) Example imaged neuron with activity in individual odor 5 trials in a single session. Dashed lines indicate odor onset and water onset. (**G**) Neuronal activity in field of view of a Drd1-Cre mouse in big airpuff and big water drop trials across days of training. Small gray dots indicate non-significantly responsive neurons. Day 0 indicates pre-training day in which no odors are presented, only water drops and airpuffs. Activity is shown in the 1 s period prior to outcome delivery, after odor onset in days 1 and 4.

The online version of this article includes the following figure supplement(s) for figure 1:

**Figure supplement 1.** Two-photon imaging of calcium activity in Drd1-Cre and Adora2A-Cre transgenic mice.

**Figure supplement 2.** Anticipatory licking in trained mice.

**Figure supplement 3.** Anticipatory licking for each of the six Drd1-Cre mice.

**Figure supplement 4.** Cell size distributions for D1 and D2 neurons.

**Figure supplement 5.** Anticipatory licking of C57BL/6 WT mice with no craniotomy/Gradient-Index (GRIN) lens implant.

nose, no outcome, small water drop, large water drop) (**Figure 1B**). Odor-outcome assignments were counterbalanced across mice and the five trial types were randomly interspersed across training sessions with 30 trials per day of each of the five odors. In each trial, the odor was presented for 1.5 s and the outcome (water or airpuff) was presented 1.3 s after odor onset. Prior to the first day of odor-outcome training, mice underwent a pre-training session (day 0) in which the airpuff and water outcomes were presented without any odors. In day 1 of odor-outcome training, mice quickly learned the odor-outcome contingencies and began licking in anticipation of water delivery in the period after odor onset prior to water delivery (**Figure 1C**). By training day 2, anticipatory licking was observed in implanted Drd1-Cre and Adora2A-Cre mice in response to rewarded odors 4 and 5, and little anticipatory licking was observed for non-rewarded odors 1–3 **Figure 1D**, **Figure 1—figure supplement 2**. The number of anticipatory licks was greater for the odor that predicted the larger reward, compared to that predicting smaller reward (2.7±0.35 vs. 2.0±0.24 on day 4 of training, p<0.0001 rank sum

test, 30 trials each odor in six mice; *Figure 1—figure supplement 3*). This suggests that the mice can perceive graded reward outcomes in addition to categorizing odors as rewarding or aversive.

## OT neuronal activity robustly and bidirectionally reflects odor-outcome contingency

D1 neurons in six Drd1-Cre mice (88±13 neurons per mouse, 529 neurons in total) and D2 neurons in six Adora2A-Cre mice (56±18 neurons per mouse, 338 neurons in total) developed both excitatory and inhibitory responses to the task odors across days of training (*Figure 1E–G*), with the most rewarded odor typically eliciting the strongest neuronal responses after training. Within the OT, there are neurons in the islands of Calleja, which express Drd1 (but not A2A). We think the inclusion of neurons from the islands of Calleja to be rare because of their characteristically small size (~8 μm diameter) and dense clustering – neither of which appear to be the case in our imaged fields of view (*Figure 1—figure supplement 4*). The anticipatory licking of the mice without a GRIN lens implant is similar to that of the implanted mice (*Figure 1—figure supplement 5*), suggesting that the implanted mice are not impaired in their ability to learn the stimulus-outcome associations.

Task-related activity of the population of D1 neurons we recorded was strikingly similar during the two aversive trial types (*Figure 2A*, columns 1–2) and similar during the two rewarded trial types (*Figure 2A*, columns 4–5) although the identity of the odorants presented during these trials varied across mice (*Figure 2—figure supplement 1*). Overall, neurons followed similar activity patterns during the two rewarded trial types and during the two aversive trial types (*Figure 2B–C*). Correlation between the activities of groups of stimulus responsive neurons in each mouse, measured by cosine similarity, was higher for same outcome trials than for opposite outcome trials after the first training day (*Figure 2D*). D1 neurons were more likely to be activated by rewarding odors than aversive odors (*Figure 2E*, 38% activated by odor 5 vs. 18% activated by odor 1 on day 4, p<0.00001, Fisher's exact test). The proportion inhibited by the same stimuli on day 4 was 18% and 13%, respectively, and not significantly different (p=0.7, Fisher's exact test). Contrary to our expectation, we found that D2 neurons were also more likely to be activated by rewarding stimuli than aversive stimuli (*Figure 2E*, 36% activated by odor 5 vs. 21% activated by odor 1 on day 4, p=0.003, Fisher's exact test; *Figure 2—figure supplement 2*). There was no significant difference between the proportion of D1 and D2 neurons activated in response to the most rewarding odor 5, on day 4 (38% and 36%, respectively, p=0.74, Fisher's exact test).

## Individual D1 neurons are more likely to encode odor valence and D2 neurons more likely to encode odor identity

By using graded odor-outcome associations, we aimed to disambiguate neuronal coding for odor valence, odor motivational salience, and odor identity. We hypothesized that, in our task, idealized valence coding neurons would exhibit similar responses to odors of similar valence (*Figure 3A*, rows 1–2), salience coding neurons would exhibit similar responses to odors of high relevance regardless of whether the outcome positive or negative (*Figure 3A*, row 3), and odor identity responsive neurons would be most likely to respond to a single individual odor (*Figure 3A*, last row). In our dataset, the prevalence of neuronal responses consistent with odor salience coding was very low (3 of 529 D1 neurons were activated for high motivational salience odors 1 and 5, but not odors 2–4, as compared to 51 neurons activated for positive valence odors 4 and 5, but not odors 1–3, p<0.000001 Fisher's test for D1 and D2 neurons). We formulated a metric to quantify salience (see Materials and methods, salience score) and found that responses consistent with salience coding were extremely rare (*Figure 3—figure supplement 1*). These data indicate that OT neuronal populations are very sensitive to the sign of the odor valence (aversive or rewarding). Thus, we focused the remainder of our analysis on odor valence or odor identity coding by D1 and D2 neurons.

We assigned each significantly responsive neuron a valence score by computing a normalized difference between the neuron's activity in response to odors of opposite valence and odors of same valence (*Figure 3B*, see Materials and methods). Identity scores were also calculated for each neuron by comparing the activity of the neuron to its most preferred stimulus to its next most preferred stimulus (see Materials and methods). Individual neurons varied widely in valence and identity scores, with some neurons having high identity scores and low valence scores (*Figure 3C*) and other neurons having high valence scores and low identity scores (*Figure 3D*). Interestingly, identity scores of D1

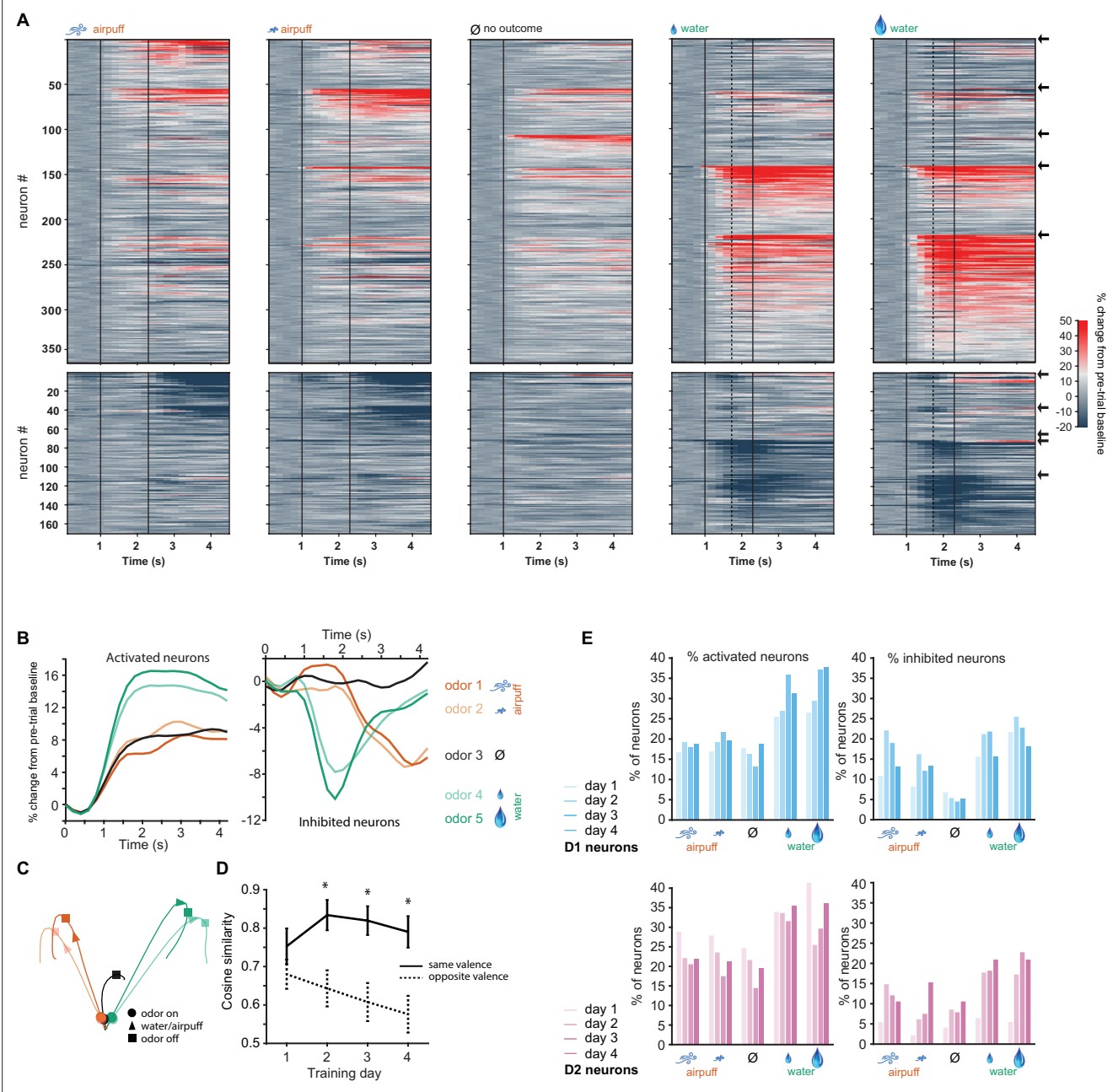

**Figure 2.** D1-type neurons in the olfactory tubercle (OT) respond most strongly to rewarded odors and respond similarly to odors of similar outcomes. (A) Activity of all activated (top) and inhibited (bottom) neurons from six Drd1-Cre mice in five trial types on day 4 of training. Vertical black lines indicate odor onset and outcome delivery time. Vertical dashed lines in the two right-most panels indicate average time of first lick by mice. Neurons are grouped by preferred stimulus, arrows on right indicate boundaries between groups. Neuronal activity in the two aversive trial types (columns 1–2) is similar and neuronal activity in the two rewarding trial types (columns 4–5) is similar. (B) Mean population activity of all activated and inhibited D1 neurons in five trial types. Odor onset at 1 s and outcome onset at 2.3 s. (C) 3D D1 neural population trajectories in five trial types. (D) Cosine similarity between groups of neurons responding to odor pairs of same valence or opposite valence (five neuronal subgroups as in A with pairwise comparisons, *p<0.05 rank sum test). (E) Proportions of significantly activated and inhibited D1 and D2 neurons on days 1–4 of training (day 4: n=529 D1 neurons, n=338 D2 neurons).

The online version of this article includes the following figure supplement(s) for figure 2:

**Figure supplement 1.** D1 neuronal activity for individual odors and mice.

**Figure supplement 2.** D2 neuronal activity in odor-outcome association task.

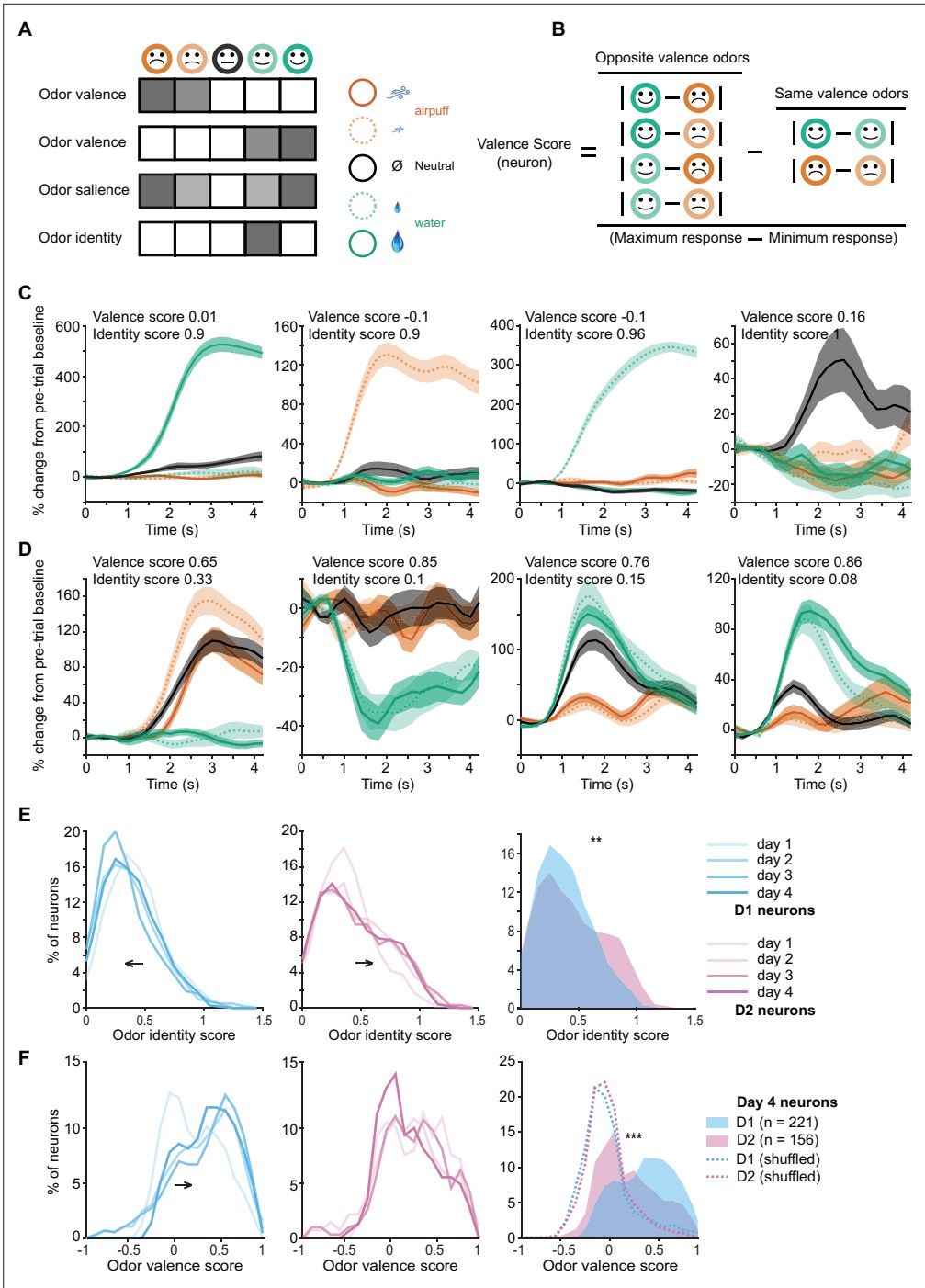

**Figure 3.** Odor valence and identity coding in individual D1 and D2 olfactory tubercle (OT) neurons develops with learning. (**A**) Hypothetical responses to the five task odors in idealized odor valence coding neurons, salience coding neurons, or odor identity coding neurons. (**B**) Calculation of valence scores for individual neurons by comparison of responses to odors of opposite valence to odors of similar valence. (**C**) Examples of four individual neurons with high odor identity coding scores and low valence coding scores. (**D**) Examples of four individual neurons with high valence coding scores and low identity coding scores. (**E**) Distributions of odor identity coding scores in D1 (blue, left) and D2 (pink, middle) neurons across 4 days of training. D1 neuron identity scores decrease and D2 neuron identity coding scores increase with training days. Right, day 4 distributions of identity scores of D1 and D2 neurons (**p<0.001, rank sum test). (**F**) Distributions of odor valence coding scores in D1 (blue, left) and D2 (pink, middle) neurons across 4 days of training. D1 neuron valence scores increase with training days. Right, day

*Figure 3 continued on next page*

*Figure 3 continued*

4 distributions of valence scores of D1 and D2 neurons (***p<0.000001, rank sum test). Dotted lines show shuffled (control) distributions where the the odor-valence pairing has been randomized.

The online version of this article includes the following figure supplement(s) for figure 3:

**Figure supplement 1.** Distributions of salience scores for D1 and D2 neurons.

neurons decreased with training (p<0.05 for day 1 vs. 4, rank sum test), while identity scores of D2 neurons trended to increase with training (*Figure 3E*, left and middle). After training, identity scores of D2 neurons were significantly higher than those of D1 neurons (*Figure 3E*, right, p<0.001 rank sum test), with more D2 neurons exhibiting clear isolated activity in response to an individual odor. In contrast, odor valence coding scores of D1 neurons significantly increased with training (p<0.0001 for day 1 vs. 4, rank sum test) while those of D2 neurons did not (*Figure 3F*, left and middle). After training, odor valence coding scores of D1 neurons were significantly higher than those of D2 neurons (*Figure 3F*, right, p<0.000001 rank sum test), and higher than that of null distributions for valence scores calculated by shuffling odor-valence assignments (dashed lines in *Figure 3F*). A shuffling procedure was not appropriate for identity scores, since they were calculated by comparing the best odor response to the next best one without regard to the associated outcome.

## D1 OT neurons encode odor valence in the absence of licking response or outcomes

In the odor-outcome association task, both odors of positive valence were accompanied by licking responses and water drop delivery, and both odors of negative valence were accompanied by airpuff delivery. As such, these trial types shared licking responses and unconditioned stimuli that likely contributed to the shared outcome-related neuronal responses we observed. In order to test whether activity of OT neurons reflected true valence coding of the odor stimulus, we constructed a probe session with two distinct training blocks. Block 1 of the session proceeded as previously described. After this first block, a 2 min break was introduced in which mice received 2 mL of free water droplets and became sated on water. Block 2 of the imaging was then conducted where mice did not lick in response to odor presentations and all outcomes (water and airpuff) were omitted (*Figure 4A*). We found that mice very rarely exhibited anticipatory licking in response to the rewarded odors in Block 2 (*Figure 4B*). Mice licked an average of 4.8±0.1 times in odor 5 trials in Block 1 and an average of 0.2±0.06 times in odor 5 trials in Block 2 (p<0.000001, rank sum test). In the following imaging data analysis, any Block 2 trials in which mice licked (5% of trials, or ~1 odor 5 trial per session) were excluded.

In Block 2, in the absence of licking and outcomes, D1 neuronal responses to odors of previously learned negative valence (*Figure 4C*, columns 1–2) were still highly similar and responses to odors of previously learned positive valence were similar (*Figure 4C*, columns 4–5), as in the full task condition (*Figure 2A*). The Euclidean distance between D1 population activities in response to odor pairs of opposite valence was 68% greater than between odor pairs of same valence. Principal components of neuronal activity and cosine similarity measures further confirmed maintenance of strong valence representation in D1 neurons in Block 2 (*Figure 4D–E*). However, valence representation was no longer present in the D2 neuronal population in Block 2 (*Figure 4E*, *Figure 4—figure supplement 1*), with the population activity for odor pairs of same valence only 12% greater in Euclidean distance (compared to 68% for D1 neurons) than for odor pairs of opposite valence.

Individual neuron valence coding scores of D1 neurons decreased significantly in Block 2 as compared to the full task condition (*Figure 4F*, rank sum test p<0.0001) indicating that the presence of the licking response, the water and airpuff outcomes, as well as the motivated thirsty state in the mouse contributed significantly to the odor-outcome-related activity of the D1 neurons. However, the mean of the distribution of the D1 valence scores in Block 2 remained above zero (0.18±0.02; p=2.8151e-13, Wilcoxon signed rank test), with many D1 neurons maintaining strong valence representation. This valence score distribution for D1 neurons in Block 2 was significantly different from a control, shuffled distribution (*Figure 4F*; p=2.1025e-12, rank sum test). D1 neurons that maintained valence coding in the absence of licking and outcome were not distinguished by their relation to licking vigor in Block 2 (*Figure 4—figure supplement 2A*). Odor valence scores of D2 neurons

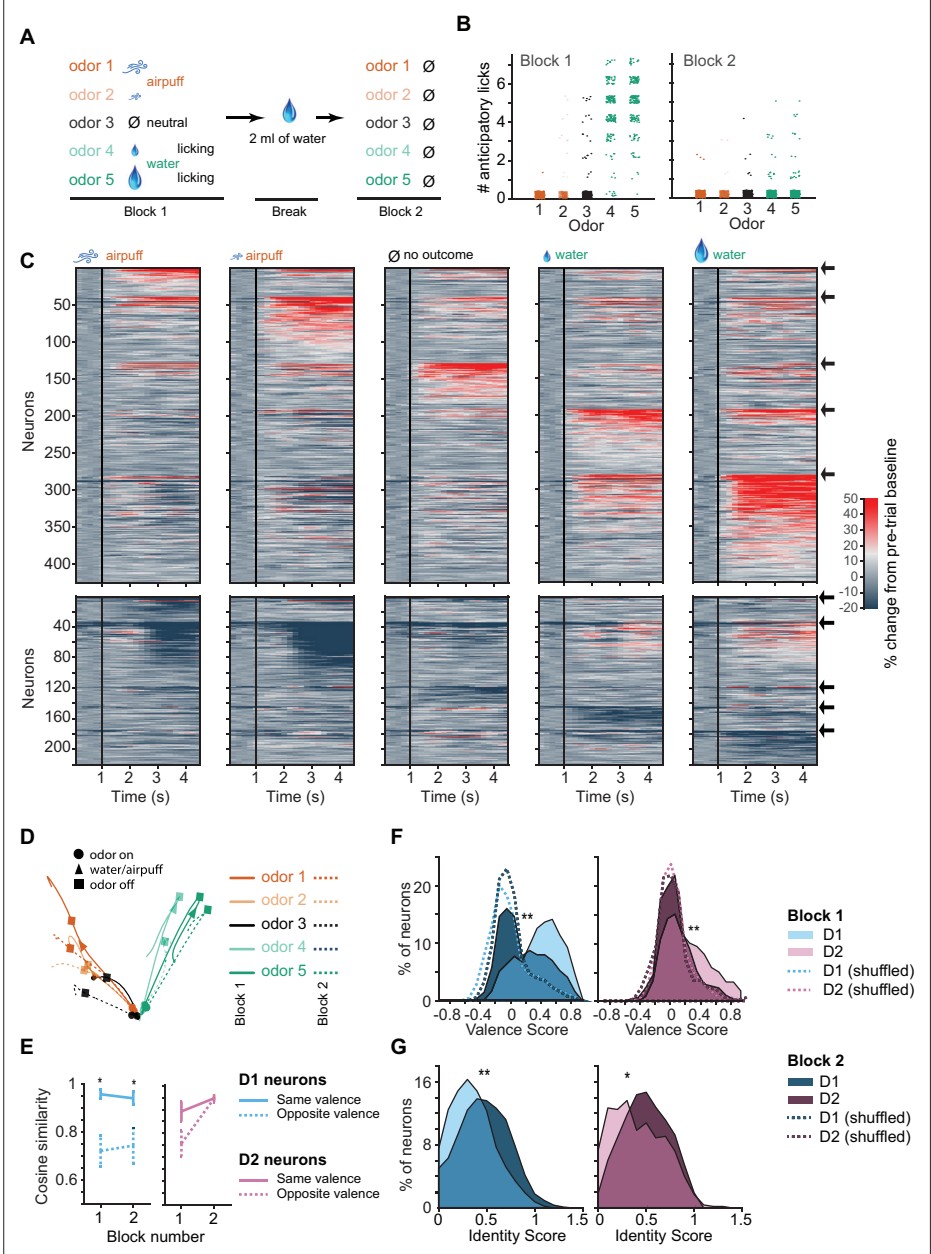

**Figure 4.** D1 neuronal valence coding is robustly preserved in the absence of licking response and absence of outcomes. (**A**) Probe session structure. In Block 2 mice are sated with water, rarely exhibiting licking, and all outcomes are omitted. (**B**) Number of anticipatory licks in response to five-odor types in Blocks 1 and 2 (n=12 mice, 16 trials each odor type in each block). Rare trials with non-zero anticipatory licks in Block 2 were omitted from following neuronal analysis. (**C**) Activity of activated (top) and inhibited (bottom) D1 neurons in response to five-odor types in Block 2. Neurons are grouped by preferred stimulus, arrows on right indicate boundaries between groups. Black vertical lines indicate odor onset. No outcome was delivered. (**D**) 3D D1 neuronal population trajectories in Blocks 1 (solid lines) and 2 (dashed lines) in five trial types. (**E**) Cosine similarity between D1 and D2 neuronal activity averages in Blocks 1 and 2 for same valence and opposite valence odors (five neuronal subgroups as in C with pairwise comparisons, *p<0.05 rank sum test). (**F**) Distributions of D1 (left) and D2 (right) neuron valence scores in blocks 1 and 2 (n=294 D1 and 177 D2 neurons, **p<0.0001, paired t-test). Dotted lines show shuffled (control) distributions where the odor-valence pairing has been randomized. (**G**) Distributions of D1 and D2 neuron identity scores. Same neurons as in panel F.

The online version of this article includes the following figure supplement(s) for figure 4:

**Figure supplement 1.** D2 neuronal activity in sated Block 2 with no licking and no water/airpuff outcomes.

*Figure 4 continued on next page*

*Figure 4 continued*

**Figure supplement 2.** Further evidence for valence coding by D1 neurons in Block 2.

**Figure supplement 3.** Neutral odor responses of valence coding neurons.

decreased in the sated condition (*Figure 4F*, right, p<0.0001 rank sum test), and the mean of the distribution of valence scores of D2 neurons in Block 2 was no longer significantly different from zero. Valence scores of D1 neurons were significantly higher than valence scores of D2 neurons in Block 2 (p<0.01 rank sum test). Concurrently, with the omission of licking and outcomes, odor identity coding scores of both D1 and D2 neurons increased in Block 2 (*Figure 4G*, p<0.01 rank sum test). Salience scores calculated for D1 and D2 neurons in both blocks were indistinguishable or lower than shuffle controls, further supporting lack of salience coding (*Figure 4—figure supplement 2B*). While most of the analysis focused on rewarding or aversive odors, we noticed that responses to neutral odors trend with aversive odor responses in Block 1, but fall intermediate between rewarded and aversive odor responses in Block 2 *Figure 3D*, *Figure 4—figure supplement 3*.

## Stimulus valence coding by D1 OT neurons is modality-sensitive

To further interrogate the stimulus valence coding property of OT neurons, the implanted Drd1-Cre mice (n=6) and Adora2A-Cre mice (n=4), which had been previously trained on the five-odor-outcome conditioning task, were then trained on a new task in which sound associations were introduced. The main motivation for these sound-odor association experiments was to determine whether sensory cues from a different modality known to activate OT neurons (*Varga and Wesson, 2013*; *Wesson and Wilson, 2010*) could become categorized with odor cues for combined valence coding. We have strong evidence from our earlier study (*Millman and Murthy, 2020*) that novel rewarded odors recruit the same OT neurons that respond to familiar rewarded odors, and therefore we wanted to see if adding a rewarding cue from a different modality will engage the same cells.

A new task was introduced in which two odor-outcome associations and two sound-outcome associations were presented. Odors 1 and 5 were paired with strong airpuff and large water reward as previously, and two sound tones (5 and 12 kHz) were used, also paired with strong airpuff and large water reward (*Figure 5A*). These sound stimuli were selected after a pilot behavioral study in a separate cohort of mice with no implants, designed to produce similar learning and anticipatory licking rates as the odor stimuli. After 1 training day with the new sound stimuli, the anticipatory licking rates of 10 mice in response to the rewarded sound stimulus were similar to that of the previously learned rewarded odor stimulus (*Figure 5B*). There was a significant difference between the number of licks for the rewarded and aversive sounds (p=8.0914e-28, two-sided Wilcoxon rank sum test), and a significant difference between the number of licks for the rewarded and aversive odors as before (p=8.2317e-54, two-sided Wilcoxon rank sum test). After 3 days of training on the new odor-sound associations, a probe session as described previously was conducted, in which mice were sated on water midway in the session and the odor and sound stimuli were then presented in Block 2 in the absence of licking or outcomes (*Figure 5A*).

In the stimulus-only condition in Block 2, we found that the same D1 neuronal populations which had previously strongly maintained similar activity patterns in response to odors of similar valence (*Figure 4C*) did not do so for odors and sounds of identical valence (*Figure 5C*). In the full-task condition when licking and outcomes (water or airpuff) were present, response similarity remained during rewarding and aversive trials (*Figure 5I*, *Figure 5—figure supplement 1*), but this similarity was lost in the stimulus-only condition. Only 5% greater Euclidean distance remained between D1 population activity in response to odor and sound of opposite valence as compared to odor and sound of same valence, unlike the 68% difference found in Block 2 of the five-odor task. Interestingly, although 9.2% of all D1 neurons (and 9% of D2 neurons) were significantly activated during at least one of the sound trial types in Block 2, these neurons were largely non-overlapping with those which responded to the odor stimuli (*Figure 5D*). We then directly compared the activity of neurons which were responsive to odors 1 and 5 in response to the matched valence odor stimuli (odors 2 and 4, in odor-odor task) and the matched valence sound stimuli (sounds 1 and 2, in odor-sound task). We found that aversive odor 1 responsive neurons became highly activated when aversive odor 2 was presented in Block 2 of the odor-odor task, but this was not the case when sound tone 1 was presented in the odor-sound task (*Figure 5E*, orange, p<0.000001 rank sum test for difference between odor-odor and odor-sound).

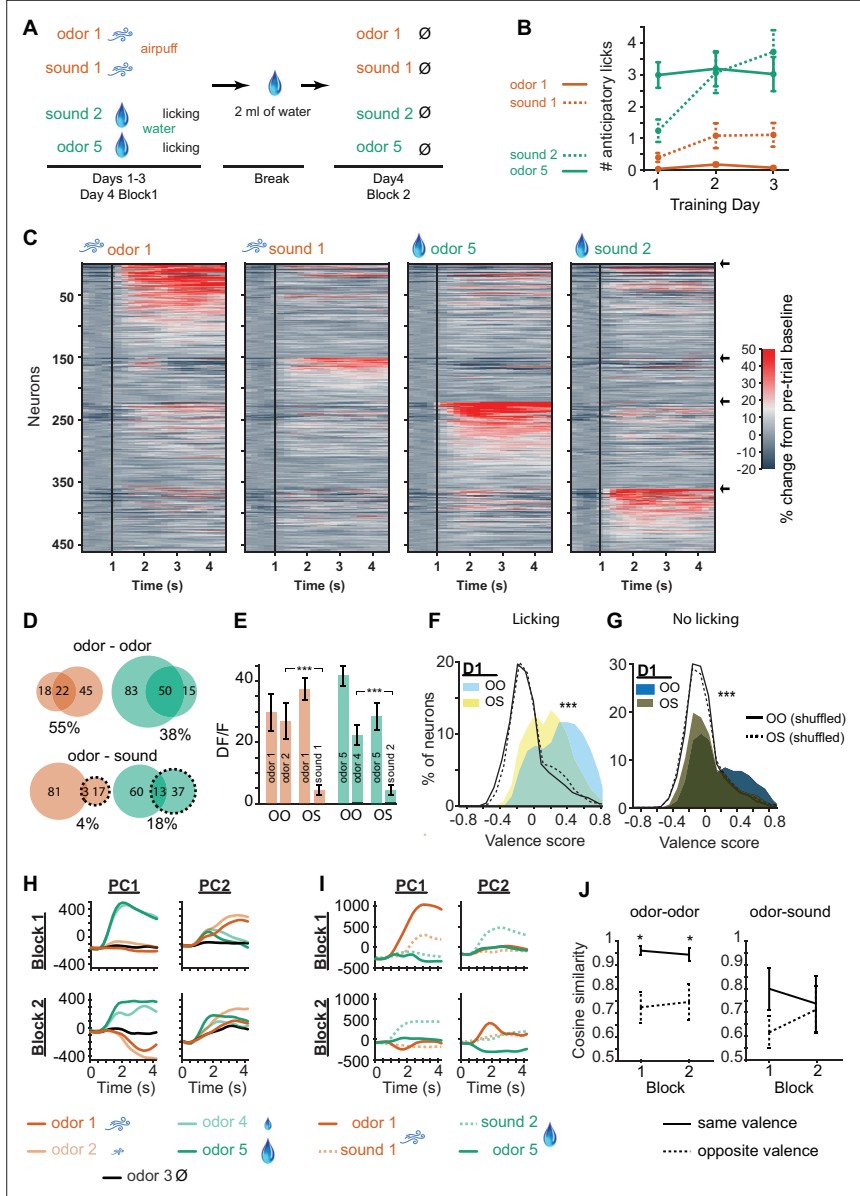

**Figure 5.** Odors and sound tones associated with identical aversive and rewarding outcomes activate different D1 neuronal subpopulations in the olfactory tubercle (OT). (**A**) Odor-sound association task structure. Three days of odor-sound training beginning at the end of the prior five-odor task training were conducted. Odors 1 and 5 associated with the strongest aversive and rewarding outcomes from previously learned five-odor task were preserved and two sound tones (5 and 12 kHz) were introduced with matching outcomes. Sound tone–outcome assignments were counterbalanced across mice. On day 4, a probe session was conducted as previously in which mice were sated prior to the second block. In the second block, mice did not exhibit licking and all outcomes were omitted. (**B**) Anticipatory licking of mice across three training days. By day 2, mice exhibit similar levels of anticipatory licking in response to the rewarded sound tone as to the rewarded odor (n=10 mice). (**C**) D1 neuronal population activity in Block 2 of day 4, in response to the learned odors and sound tones, and in the absence of licking, airpuffs, or water delivery. Neurons are grouped by preferred stimulus, arrows on right indicate boundaries between groups. Unlike in *Figure 4C*, distinct sets of D1 neurons are activated in response to odors and sound tones associated with identical outcomes. (**D**) Overlap in neurons responding to stimuli predicting similar aversive (orange) and rewarding (green) outcomes in the sated condition in the odor-odor task (top) and odor-sound task (bottom). Top orange, numbers of neurons in Block 2 of the odor-odor task that were activated in response to either aversive odor 1, aversive odor 2, or both of them (overlap), with % overlap indicated below. Top green, same for rewarding odors 5 and 4. Bottom, same for Block 2 of the odor-sound task (sounds in dashed lines). (**E**) Orange, mean activity of odor 1 activated neurons to the corresponding matched valence odor stimulus in Block 2 of

*Figure 5 continued on next page*

*Figure 5 continued*

the odor-odor (OO) task and matched valence sound stimulus in Block 2 of the odor-sound (OS) task (n=50 and 101 neurons). Green, same for odor 5 responsive neurons (n=100 and 134). (**F**) Distributions of valence scores of D1 neurons in the odor-odor task and odor-sound task in the full task condition with licking and outcomes. Shuffle control distributions are also shown as lines. (**G**) Same for stimulus-only condition in Block 2. (**H**) Principal components of D1 neuronal population activity in Blocks 1 and 2 in odor-odor task. (**I**) Same for odor-sound task. (**J**) Cosine similarity in the neuronal trajectories of stimulus responsive neuronal groups in the odor-odor task as compared to the odor-sound task in same valence vs. opposite valence stimulus pairs (four neuronal subgroups as in C with pairwise comparisons, *p<0.05 rank sum test).

The online version of this article includes the following figure supplement(s) for figure 5:

**Figure supplement 1.** D1 neuronal activity in full odor-sound task, including licking and water/airpuff outcomes.

**Figure supplement 2.** D2 neuronal activity in full odor-sound task, including licking and water/airpuff outcomes.

**Figure supplement 3.** D2 neuronal activity in sated Block 2 of odor-sound task with no licking and no water/airpuff outcomes.

---

Similarly, rewarding odor 5 responsive neurons became highly activated when rewarding odor 4 was presented in Block 2 of the odor-odor task, but this was not the case when rewarding sound tone 2 was presented (*Figure 5E*, green, p<0.000001 rank sum test). We also calculated valence scores for these neurons in the odor-odor task and the odor-sound task. In the standard task condition with licking and outcomes (*Figure 5F*), as well as in the sated condition with no licking or outcomes (Block 2, *Figure 5G*), the valence scores of D1 neurons in the odor-sound task were significantly lower than the valence scores of the neurons in the odor-odor task (p<0.0001, rank sum test). This occurred despite the presence of similar levels of anticipatory licking in response to the rewarded sound tone, and even though the odor- and sound-related outcomes in the odor-sound task were identically matched while those in the odor-odor task were not.

Finally, we compared the principal components of D1 neuronal population activity in response to the five odors in Blocks 1 and 2 in the odor-odor task and activity in response to the odor-sound stimuli in Blocks 1 and 2 in the odor-sound task. Principal components of D1 neuronal populations were very similar in odors predicting similar outcomes in the typical task condition (*Figure 5H*, top) and the sated condition with no licking or outcomes (*Figure 5H*, bottom). However, there was no such clear similarity of D1 neuronal population principal components in response to odors and sound tones predicting identical outcomes (*Figure 5I*), suggesting distinct neuronal activity trajectories in response to each of the four stimuli used in the odor-sound task. Measurement of the cosine similarity in the two conditions confirmed this interpretation (*Figure 5J*). D2 neurons in Block 2 of the odor-sound task did not display significant valence coding (*Figure 5—figure supplements 2 and 3*), which is not surprising given the lack of valence coding in Block 2 of the five-odor task (*Figure 4G*).

## Discussion

The results of these experiments offer key insights into the function of OT neuronal circuitry during odor association learning. We find some surprising similarities and some key differences in the neuronal responsivity of D1- and D2-type OT neurons, in the first single neuron imaging experiments of specific neuronal subtypes in the OT. We found that D1 neuronal populations clearly responded to both positive and negative odor valence. This is the first demonstration of bidirectional odor valence coding by OT neurons, as no previous real-time recordings of OT neurons in response to aversive valence odors have been reported. We therefore can conclude that the OT is likely to be involved in learning about both positive and negative odor associations, rather than the alternative possibilities that the OT is only involved in learning rewarded odor associations, or that it encodes odor salience rather than signed odor valence. We note that a higher proportion of neurons was active in response to the rewarded odors than the aversive odors; however, this may be an effect of possible high value of the water reward outcome in water restricted mice as compared to the aversiveness of a relatively harmless airpuff to the nose. Surprisingly, similar proportions of D1 and D2 neurons responded to the five task odors, challenging our initial hypothesis that D2 neurons may respond more strongly to negatively reinforced odors, based on immediate early gene expression results (*Murata et al., 2015*; *Murata et al., 2019*). This suggests that models of dopamine acting to potentiate responses

of D1R expressing neurons and de-potentiate responses of D2R expressing neurons or dopamine dips potentiating D2R expressing neurons (*Bamford et al., 2018*; *Iino et al., 2020*; *Surmeier et al., 2007*; *Yagishita et al., 2014*) are not sufficient to account for the neuronal activity we observed, in accordance with a more complex interplay between dopamine and the two striatal neuronal types (*de Jong et al., 2019*; *Kutlu et al., 2021*). Other inputs to D1- and D2-type neurons in the OT, including strong inputs from piriform cortex (*White et al., 2019*), could also create differentiated activity in these neurons.

We then addressed a long-standing question about the factors driving the responses of OT activity in odor-outcome behavioral paradigms. In previous OT recording experiments, go/no-go tasks were used where licking in response to the rewarded odors was required. In our classical conditioning task, mice similarly exhibited anticipatory licking at the onset of the rewarded odors, although the water delivery was not contingent on this response. In all of these conditions, the odor presentation itself was in every case coupled with the motor licking action of the mouse. Thus, it was unclear whether the neuronal activity recorded occurred as a result of the odor stimulus or the licking response. Previously, it was reported that the onset of the recorded neuronal responses preceded the onset of the licking action by ~200 ms (*Gadziola et al., 2015*; *Millman and Murthy, 2020*); however, this time lag is well within the time range typically seen in motor preparatory neuronal activity (*Svoboda and Li, 2018*; *Tanaka et al., 2021*) and could still be linked to the licking action or its preparation. To address this confound, we sated the mice on water and omitted the outcomes associated with the odors. We observed that in 95% of the trials in this condition, mice exhibited zero licks and we analyzed neuronal activity from only these trials. We observed that D1 OT neurons continued to respond to the learned odors and continued to strongly differentiate odor valence in the absence of licking or outcomes. This finding indicates that the critical sensory transformation step of integrating odor identity information and outcome information takes place in the OT D1 neurons, and that the odor-outcome-related activity seen in our recordings and previous recordings are not the result of unintended correlations with movement. This is especially notable given the low motivational state of the sated mouse in this experimental condition, suggesting that D1 OT neurons, at least temporarily, maintain odor valence memory even in conditions when the outcomes associated with the learned odors become less consequential. It should be noted, however, that in the condition where licking and water and airpuff delivery were present, the odor-outcome representation in D1 and D2 neurons was enhanced as compared to the stimulus-only sated condition. This indicates enhanced OT engagement during conditions in which the mice were motivated and behaving, consistent with other reports of increased neuronal activity modulation in sensory regions in attentive behaving mice as compared to passive stimulus presentation conditions (*Busse et al., 2017*; *Carlson et al., 2018*; *Pakan et al., 2018*).

We also note that D1 neurons are heterogeneous in terms of their valence coding, and a sizable fraction loses its valence coding in the absence of licking and outcome (Block 2). We did not find distinct populations of neurons responding to licking and valence separately (*Figure 4—figure supplement 2A*), indicating that licking and valence responsivity are intertwined factors. The heterogeneity of D1 responses could arise from natural stochasticity in cellular plasticity mechanisms involved in the generation of the firing patterns such that not all neurons (and connections) are modified in the desired direction. Our observed heterogeneity could, in part, be due to unintended labeling of additional neuronal types. It is unlikely that we imaged ventral pallidal neurons since we positioned our GRIN lens as ventrally as possible, confirmed the location histologically excluding mice in which we suspected more dorsal localization, and since expression of Drd1 and A2A is very low in the ventral pallidum (*Figure 1—figure supplement 4A*, Allen Brain Atlas). As discussed in the Results (*Figure 1—figure supplement 4*), it is unlikely that a significant number of neurons from the islands of Calleja were included in our analysis.

Our findings on the activity of OT D2 neurons provide an interesting contrast to the robust valence coding property of D1 neurons. While D2 neuronal populations differentiated between rewarded and aversive outcomes in the task condition involving licking and outcomes, this representation was significantly weaker than that of D1 neurons and disappeared in the stimulus-only condition when licking and outcomes were omitted. This result contrasts with photometry data in which reward contingency information was not observed in average D2 population activity in a go/no-go task (*Gadziola et al., 2020*), suggesting it is possible to differentiate rewarded trials in non-averaged D2 neuronal activity in a condition when the licking response is present. However, our data does demonstrate that reward

contingency information is more readily represented by D1 neurons than D2 neurons. Individual D2 neurons were much more likely to respond to an individual odor out of the five task odors and exhibit little responsivity to any of the other four odors, suggesting odor-identity rather than odor-valence responsivity. Increased training resulted in increases in the valence-coding scores of D1 neurons and decreases in their odor-identity coding scores, while the opposite pattern was observed in D2 neurons, with increased odor-identity coding scores with training.

How D1 and D2 neuronal representations of learned odors diverge based on inputs to these neurons and the effect of dopamine onto them, and how they then subsequently shape behavior as they influence downstream targets are both important areas to investigate. Cre expressing neurons in the dorsal striatum from the Drd1-Cre and Adora2A-Cre mouse lines used in our imaging experiments have been previously shown to project via the distinct direct and indirect striatal output pathways (*Gerfen et al., 2013*), and as such have opposing overall effects on thalamus and cortex, potentiating behavioral output and inhibiting it, respectively. There is much less known about the differential projections of D1 and D2 neurons in the OT (*Heimer et al., 1987*; *Zhou et al., 2003*) and their function (*Gadziola et al., 2020*; *Murata et al., 2015*; *Murata et al., 2019*). Given the relatively weak odor-outcome representation by OT D2 neurons, the question also remains whether these neurons play a significant role in odor-outcome association learning, or whether D1 OT neurons predominantly contribute to this function.

Informed by previous studies showing that sound cues can also activate OT neurons (*Varga and Wesson, 2013*; *Wesson and Wilson, 2010*), we asked whether individual neurons respond similarly to odor and sound cues that predict the same reward or airpuff, which would point to multimodal integration within the OT to form a unified valence representation. Such a result might also be expected if neuronal responses were closely related to licking response and outcome, since the odor and sound cue lead to the same learned motor behavior. Consistent with previous results, we found that 24% of D1 neurons and 14% of D2 neurons were activated in response to at least one of the sound tones in the full task condition with licking and outcomes. However, unlike the large overlap and similarity between neuronal activity in response to different odors predicting similar outcomes, we found little overlap between neurons responding to odors and sounds predicting identical outcomes. This finding points to two conclusions.

First, corroborating our previous result, we can conclude that the odor valence-related OT activity was not primarily a result of the licking response of the mouse. We demonstrate that behaviorally, the anticipatory licking rates of the mice in response to the rewarded odor stimulus and the rewarded sound stimulus are similarly high after training, yet stimulus-outcome representation in the OT during the odor-sound task is minimal in the period prior to outcome delivery even in the presence of matched anticipatory licking rates. Second, we can conclude that stimulus valence representation by D1 OT neurons does not automatically generalize to multimodal stimuli. Our previous study (*Millman and Murthy, 2020*) demonstrated that introducing novel rewarded odors recruits the same valence cells that respond to familiar rewarded odors. Our current data indicate that adding a new stimulus of a different modality does NOT automatically recruit the same reward category cells. Sound-related responses in OT do occur and supra-additive effects of odor and sound stimuli have been reported (*Wesson and Wilson, 2010*). While there were clearly strong neuronal responses to sound tones in our experiments, our design did not include multiple sound tones for outcomes of the same valence as we did for odors in the original five-odor task. Therefore, we cannot be sure that a cell responding to a rewarding tone is simply because of sensory tuning, or because of valence coding. Future studies can address this important issue, since it will inform us about the possible specialized role of limbic brain regions such as the OT to store emotionally charged odor memories, a property that may be unique to the sensory modality of olfaction.

An unavoidable consequence of GRIN lens imaging of deep brain regions is tissue damage, but we are confident that our key conclusions are not affected by this issue. We made concerted efforts to minimize the damage caused by the implanted cannula. First, the cannula was constructed from highly biocompatible and thin-walled polyamide tubing and quartz floor which have previously been shown to minimize glial scar tissue (*Bocarsly et al., 2015*). Second, 2 mm of the cortex were removed by suction prior to the virus injection and cannula implantation to minimize pressure in the brain. Finally, the mice were allowed to recover for at least 1 month prior to the onset of behavioral training. After the extensive recovery period, mice learned the association tasks rapidly. The anticipatory licking of

the mice with no GRIN lens implant is similar to that of the implanted mice (*Figure 1—figure supplement 5*), suggesting that the implanted mice are not impaired in their ability to learn the stimulus-outcome associations.

The main comparisons we make in our study are likely not a function of the damage caused by the lens. First, the differences in the valence coding of D1 and D2 neurons are unlikely to be a result of lens damage, since there is no reason to suspect that damage caused by the lens will differentially affect D1 and D2 neuronal activity in the OT. D1- and D2-type neurons in the OT and the striatum typically receive inputs from similar upstream structures and their inputs were not generally altered by the lens damage. Second, there is no reason to suspect that the robust valence coding in the absence of the licking and outcomes by D1 neurons could be a result of the lens damage. Third, there is no particular reason to suggest that the distinction between the odor and sound responsive neurons in the last set of experiments would be a result of the damage caused by the lens as the possible auditory cortical projections to the OT arrived from the posterior direction. Finally, the nature of the valence-related responses we observe in the OT are similar to those observed by others using tetrode recordings (*Gadziola et al., 2015*; *Millman and Murthy, 2020*) in which there was presumably less damage to the striatum.

In summary, we find that D1 OT neurons are more likely to encode learned odor valence than D2 neurons, and conversely less likely to encode odor identity. We also demonstrate, for the first time, that even when the licking response to rewarded odors is eliminated, OT D1 neurons continue to robustly encode odor valence suggesting that this stimulus to valence transformation by the OT precedes the motor action itself. These results also suggest that OT odor valence representation could inform downstream brain regions of the value of odor stimuli. Finally, we find that stimulus valence representation by OT neurons is limited to olfactory stimuli suggesting a specialized role of the OT in assigning emotional tags to odors based on previous experience. Further investigation into the relative contributions of D1 and D2 OT neurons to odor association learning and the neural mechanisms that result in the differential responses of these neuronal types is required. These neuronal imaging results suggest a specialized role for the OT in odor valence memory, and further studies can be conducted to assess the causal contribution of the OT to the hypothesized unique emotional qualities of olfactory memory.

# Materials and methods

## Animals

Adult male and female heterozygous B6.FVB(Cg)-Tg(Drd1-Cre)EY262Gsat/Mmucd and B6.FVB(Cg)-Tg(Adora2A-Cre)KG139Gsat/Mmucd (MMRRC) mice were 2–6 months of age at the start of the experiments. Due to the highly consistent colocalization of A2A receptors and D2 dopamine receptors in the striatum, no colocalization of A2A receptors with D1 dopamine receptors (*Gerfen and Surmeier, 2011*; *Svenningsson et al., 1998*), and previously established use of Adora2A-Cre mice for indirect pathway specific manipulation (*Cui et al., 2013*), we proceeded with the use of Adora2A-Cre mice to image D2-type neurons. All experiments were conducted with approved protocols and in accordance with Harvard University Animal Care Guidelines.

## Cannula assembly

Custom-designed cannula were assembled in house. 6.2 mm length 1.1 mm diameter ultra-thin wall biocompatible polyimide tubing (MicroLumen) which was demonstrated to cause minimal inflammatory response in the brain (*Bocarsly et al., 2015*) was used for the walls of the cannula. 150 µm thickness quartz coverslips (Electron Microscopy Sciences) were cut to 1 mm diameter disks in a laser cutter and used as the floor of the cannula. Cut quartz disks were held with the assistance of vacuum under the view of a surgical microscope, attached to the polyimide tube with Norland Optical Adhesive NOA 68 (Edmund Optics), and adhesive was cured with UV light source (ThorLabs). Directly prior to surgical implantation, cannula were inspected and disinfected with the use of the UV light source.

## Surgery

Naïve mice underwent surgical virus injection, cannula implantation, and head fixation plate implantation, prior to any behavioral training. Mice were anesthetized with an intraperitoneal injection of

a mixture of xylazine (10 mg/kg) and ketamine (80 mg/kg) and placed in a stereotaxic apparatus. A 1.4 mm craniotomy was performed at 1.5 mm AP, 1.3 mm ML in the right hemisphere. A 22 G needle was used to suction 2 mm below the brain surface prior to virus injection. A pulled glass micropipette attached to a nanoinjector (MO-10, Narishige) was used to inject 400 nL of pGP-AAV9-syn-FLEX-jGCaMP7s-WPRE (Addgene) virus at a depth of 4.8 mm DV at a rate of 100 nL/min. Five minutes after the injection was completed, the glass pipette was raised out of the brain over the course of another 5 min. The sanitized cannula was then held lightly with a dental paper point inserted into its center and attached to the stereotaxic arm. The cannula was lowered over the course of 10 min to a depth of 4.9 mm DV. The cannula was secured onto the skull with cyanoacrylate glue and a head fixation plate was also glued to the skull behind the cannula. Dental cement (MetaBond) was then used to cover the skull and headplate. The opening of the cannula was covered with a silicone sealant (KwikSil). Mice were single housed after surgery. After a period of 4 weeks to allow for virus expression and the reduction of the inflammatory response to the insertion of the cannula, a 1 mm diameter, 3.4 mm long 0.5NA GRIN lens (ThorLabs) was inserted into the cannula and behavioral training and imaging begun.

## Behavioral training

Mice were water restricted to reach 85–90% of their initial body weight and provided approximately 1–1.5 mL water per day in order to maintain desired weight. Mice were habituated to head fixation and drinking from water spout prior to initial training session. In the pre-training (day 0), mice were provided large water drops (20 µL), small water drops (10 µL), strong airpuff (10 PSI), and weak airpuff (5 PSI) in identical trial structure as full five-odor conditioning task, but odors were not used. Each mouse was then assigned five odor-outcome contingencies with the monomolecular odorants hexanal, limonene, anisole, eucalyptol, and heptanal. Odors were delivered via a custom-built olfactometer as described previously (*Soucy et al., 2009*) with a 1.5 L/min flow rate at a concentration of 20% for 1.5 s each. Odor-outcome contingencies were assigned so as to equally or close to equally counterbalance the odor-outcome contingencies across each cohort of mice. The delay between the solenoid valve opening (which is denoted as time 0 throughout the manuscript) and the arrival of the odor at the mouse's nostrils was measured with a miniPID instrument (Aurora Scientific, Inc).

In days 1–4 of training, each of the five odors and associated outcomes were provided 30 times with 20 s inter-trial intervals. In 10% of trials (three trials of each trial type), the outcomes were omitted; however, this number of trials was not later found to be sufficient for the analysis of neuronal activity. Trial types were interspersed randomly across the session, with the constraint that equal numbers of each trial type occurred in the first and second half of each session to ensure equal trial type representation for the duration of each imaging session. Licking of the water delivery spout was measured throughout training and imaging with the use of a capacitance sensing Arduino circuit. Behavioral events control and recording was conducted with Python with adapted use of the ScopeFoundry platform (http://www.scopefoundry.org/) and National Instruments DAQ hardware.

Prior to Block 2 in the probe session in day 5 of training, 2 mL of free water was provided to the headfixed mouse over the course of 2 min. Fifteen trials of each trial type were then presented in identical task structure as in Block 1, however all water and airpuff outcomes were omitted.

Training in the odor-sound task occurred 1–2 days after the completion of the five-odor task training. Odors 1 and 5 with strong airpuff and large water drop outcomes were preserved, and 5 and 12 kHz sound tones were introduced paired with the same strong airpuff and large water drop outcomes. As in previous task, 30 trials of each condition were randomly interspersed across session time. Onset and duration (1.5 s) of the sound stimuli and associated outcomes was identical to that of the odor stimuli. After 3 days of odor-sound task training, a probe session with two blocks was conducted as in the original five-odor task.

## Two-photon imaging of calcium activity

A custom-built microscope was used for in vivo imaging as described previously (*Kapoor et al., 2016*; *Petzold et al., 2009*). Imaging was conducted at 5 Hz with an air objective (10×, Leica) at 930 nm using a Ti:sapphire laser (Chameleon Ultra, Coherent) with a 140 fs pulse width and 80 MHz repetition rate. Image acquisition, scanning, and stimulus delivery were controlled by custom-written software (*Kapoor, 2022*; deposited in GitHub) in LabVIEW (National Instruments). Prior to two-photon imaging, the position of the GRIN lens and approximate neuronal imaging plane was determined with

a camera. The head fixation plate was mounted on an adjustable pitch and roll platform (ThorLabs) which allowed for manual adjustment of the lens angle to parallel alignment with the objective. The depth of the imaging plane was adjusted each day to closely match that of the previous imaging days, capturing the same or highly overlapping neuronal populations across days of training.

## Calcium imaging data analysis

Imaging data was motion corrected with Non-Rigid Motion Correction (NoRMCorre) (*Pnevmatikakis and Giovannucci, 2017*). The activity of single neurons was then isolated and background subtracted with the use of CalmAn (*Giovannucci et al., 2019*) followed by manual refinement. The number of putative calcium transient events in each neuron was then quantified based on a criterion of activity 3 standard deviations above a temporally proximal baseline lasting longer than 5 frames, and neurons with less than two recorded transients were not used in the analysis. Due to baseline fluorescence fluctuations in single neurons, the activity of each neuron in individual trials was normalized to a 1 s pre-trial baseline to isolate trial event-related fluorescence changes. Due to the 5 Hz imaging rate and following use of a minimal locally weighed smoothing filter to reduce noise in recordings, in some cases event-triggered changes in fluorescence may appear to begin in 1–2 frames prior to the event time.

$$Valence\ Score = \frac{mean\ (differences\ between\ opposite\ valence\ odors) - mean\ (differences\ between\ same\ valence\ odors)}{|maximum\ response - minimum\ response|}$$

$$Identity\ Score = \frac{Strongest\ odor\ response - second\ strongest\ odor\ response}{strongest\ odor\ response}$$

$$Salience\ Score = \frac{mean\ (differences\ between\ different\ salience\ odors) - mean\ (differences\ between\ similar\ salience\ odors)}{|maximum\ response - minimum\ response|}$$

Odor valence coding scores and odor identity coding scores were computed for neurons which had significant mean activity deviations of >30% from pre-trial baseline as shown below. Due to the variability in the neutral odor responses which often tracked closely with either the aversive odors or the rewarded odors in neurons which clearly differentiated between the rewarded and aversive odors, the neutral odor responses were not included in the numerator of the valence score formula (but they could still feature in the denominator if neutral odors caused maximum or minimum response for a given neuron). A salience score was calculated analogous to the valence score using the formula noted above. Distributions of odor valence scores and odor identity scores across neuronal types and task conditions were compared with the non-parametric Wilcoxon rank sum test. We generated null distributions by shuffling the odor-valence pairing. Since the identity scores were calculated by comparing the best odor response to the next best one, the shuffling procedure does not affect the identity scores and was not done for those distributions. Matrix difference measures of neuronal population activity were obtained by taking the Euclidean norm of the difference between population activity in response to pairs of stimuli used in the task. The mean norm for stimuli pairs of opposite valence was then compared to the mean norm for stimuli pairs of the same valence. Neuronal population activity dimensionality reduction and trajectory analysis was conducted with the use of the DataHigh toolbox (*Cowley et al., 2013*) with all D1 and D2 neurons collected in the dataset and trial types used as input conditions.

## Histological confirmation of imaging site

After completion of imaging experiments, mice were transcardially perfused, and the brains were removed from the skull. Coronal floating sections were cut using a vibratome (Leica VT1000S). Brain sections were imaged using the Zeiss Axio Scan slide scanner at the Harvard Center for Biological Imaging to visualize the location of GCaMP expression and the location of cannula tip. Brain section images were matched and overlaid with the Paxinos and Franklin Mouse Brain Atlas cross-sections to identify imaging location. Six of eleven implanted Drd1-Cre mice had confirmed OT imaging locations, while in others the tip of the cannula was located in ventral pallidum or nucleus accumbens. Eight of eleven implanted Adora2A-Cre mice had confirmed OT imaging locations, but only six of eight produced satisfactory imaging results. Only mice with confirmed OT imaging locations and successful imaging results were used in later analysis.

## Statistical analysis

We primarily used the non-parametric Wilcoxon rank sum test (referred to simply as rank sum test in the main text) and Fisher's exact test, since we could not assume normally distributed data in most cases. Comparisons involving t-tests were paired and two-tailed.

## Acknowledgements

We thank Dr Naoshige Uchida and Dr Mitsuko Uchida for their helpful guidance throughout this project and comments on the manuscript. We thank Dr Hao Wu for his help in establishing the Python-based, ScopeFoundry behavioral control system for the experiments. We also thank Selina Qian and Rebecca Fisher, for helping with animal colony maintenance and habituation, and helpful discussions. This work was supported by grants from the NIH (R01DC017311, R01NS116593, F32DC017891) and a Bipolar Disorder Seed Grant from the Harvard Brain Initiative.

## Additional information

### Funding

| Funder | Grant reference number | Author |
|---|---|---|
| National Institutes of Health | F32DC017891 | Nuné Martiros |
| National Institutes of Health | R01DC017311 | Venkatesh N Murthy |
| National Institutes of Health | R01NS116593 | Venkatesh N Murthy |
| Harvard Brain Science Initiative | Bipolar Disorder Seed Grant | Venkatesh N Murthy |

The funders had no role in study design, data collection and interpretation, or the decision to submit the work for publication.

### Author contributions

Nuné Martiros, Conceptualization, Data curation, Formal analysis, Funding acquisition, Investigation, Writing – original draft; Vikrant Kapoor, Investigation, Methodology, Visualization, Writing - review and editing; Spencer E Kim, Investigation, Methodology; Venkatesh N Murthy, Conceptualization, Funding acquisition, Project administration, Resources, Supervision, Writing - review and editing

### Author ORCIDs

Nuné Martiros (iD) http://orcid.org/0000-0001-9197-7711
Vikrant Kapoor (iD) http://orcid.org/0000-0002-0185-3836
Spencer E Kim (iD) http://orcid.org/0000-0003-3007-4417
Venkatesh N Murthy (iD) http://orcid.org/0000-0003-2443-4252

### Ethics

All experiments were conducted with approved protocols and in accordance with Harvard University Animal Care Guidelines. All of the animals were handled according to approved institutional animal care and use committee (IACUC) protocols (#29-20) of Harvard University.

### Decision letter and Author response

Decision letter https://doi.org/10.7554/eLife.75463.sa1
Author response https://doi.org/10.7554/eLife.75463.sa2

## Additional files

### Supplementary files

• Transparent reporting form

## Data availability

Raw data, source data, metadata, and analysis code have been uploaded into Dryad linked to this manuscript.

The following dataset was generated:

| Author(s) | Year | Dataset title | Dataset URL | Database and Identifier |
|---|---|---|---|---|
| Martiros N, Murthy V, Kapoor V, Kim S | 2021 | Two-photon imaging of D1 and D2 type neurons in the olfactory tubercle of behaving mice | https://doi.org/10.5061/dryad.6hdr7sr28 | Dryad Digital Repository, 10.5061/dryad.6hdr7sr28 |

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
