## [Editor Report]

Martiros et al. monitored the activity of dopamine D1 and D2-expressing neurons within the ventral striatum olfactory tubercle using 2-photon microscopy, as mice learned to associate odors and tones with either positive or aversive outcomes. Authors find differential roles for these neurons in the learning of odor valence and tone outcome associations. Overall, this study was deemed as an interesting and important contribution to our understanding of the neural basis of cue encoding.

---

## [Decision Letter]

**Decision letter after peer review:**

Thank you for submitting your article "Distinct representation of cue-outcome association by D1 and D2 neurons in the olfactory striatum" for consideration by *eLife*. Your article has been reviewed by 3 peer reviewers, and the evaluation has been overseen by a Reviewing Editor and Michael Frank as the Senior Editor. The following individual involved in review of your submission has agreed to reveal their identity: Daniel W Wesson (Reviewer #1).

Martiros et al. monitored the activity of dopamine D1 and D2-expressing neurons within the ventral striatum olfactory tubercule (OT) using 2-photon microscopy, as mice learned to associate odors and tones with either positive or aversive outcomes. D1 neurons robustly encoded learned odor valence, and D1 neurons maintained an odor valence representation even when appetitive or aversive unconditioned stimuli were removed. In contrast, D2 neurons were more selective to odor identity than D1 neurons, and odor valence coding in D2 neurons was dependent on odor outcome associations. Finally, authors show that in D1 neurons, odor and tone outcome associations recruit largely non-overlapping neuronal ensembles. Overall, this study was deemed by reviewers as an interesting and important contribution, however, they did identify a need for added details about data analyses and statistical comparisons for a clearer interpretation.

Essential revisions:

1) Authors should provide more details about the specific cell types being monitored in the D1 and A2A-cre mice since cre is present in neurons, not just medium spiny / striatal projection neurons. The authors did not exclude these other cells in their analysis or experimentally so any revision should be clear about inclusion of other types of neurons (e.g., D3).

2) The function of the different subpopulations of 'valence coding' D1 neurons should be clarified in both the Results and Discussion. Relatedly, it is unclear how neuronal responses to neutral odors are taken into consideration when calculating valence scores. It is suggested to include the distributions of valence scores with shuffled cell-odor pairs as reference points. Statistical tests for comparing odor valence and identity distributions should be described in more detail.

3) It is unclear how odor identity (and sound frequency) is represented in neuronal response patterns. The authors should show neuronal responses grouped by mice / odorant-unconditioned stimulus combinations, in addition to showing neuronal responses grouped by the trial outcomes only.

4) There was concern about a large portion of the brain damaged by the placement of the GRIN lens. The implication of this damage needs to be discussed.

5) It is suggested to better link behavior to neural responses- specifically the timing of licking with the GCaMP response.

6) Authors should provide a clearer rationale for the tone learning study. The motivation for the odor-tone association experiments is unclear. The main result is that different subpopulations of OT neurons encode odor and tone associations. However, this does not appear to be a question the authors set out to address a priori.

*Reviewer #1 (Recommendations for the authors):*

This study is a much needed addition to the field, and the manuscript is excellent and well done. I have a few number of major and a longer list of minor issues I'd appreciate being addressed.

Figure 4F2 – the authors package these results that most D1 neurons maintain valence coding in absence of instrumental responding or conditioned outcomes, and while this is true, a good amount do not. It would be beneficial for the authors to elaborate upon what those two different subpopulations of D1 neurons are doing both in Results and Discussion.

The authors do show that some cells display auditory valence coding, yet conclude across the population there is no such activity. It would be helpful for the authors to describe activity of the cells which do in fact show learned responses to the tones. Related to this, in some places the authors describe their results as "not modality invariant". Perhaps this could be rephrased to read, "modality variant" to avoid double negatives. Finally, isn't the finding that there are responses to multisensory stimuli and that these responses are dependent upon learning reason to reconsider titling this paper as originating from the "olfactory striatum"? These results show this is not simply an "olfactory" structure, and as described in Wesson 2020 (Tubular Striatum), the concept of calling the olfactory tubercle the "olfactory striatum" lost ground decades ago. Of course, I'm all for scholarly creativity of the authors and am not asking them to change the title, but more to think about what message this sends to readers.

Overall, it would be helpful to see the timing of licking, even if on average, to help relate the gcamp response to behavior better. This is important since the authors used a fixed ITI of 20s and normalized each neuron's activity to a 1s pre-trial baseline. But can we be sure there was no anticipatory-related activity during that time? More specifically, it would be nice to show a new figure/inset of Figure 1C, day 1 and/or 4 in raster-format (similar to 4B perhaps) so readers can gain an appreciation for the timing of the licks specifically in this panel, if possible (it's reported in the first results paragraph that mice began licking after odor onset before water delivery, but it is not shown as such).

I'd appreciate the authors elaborating upon the argument from their Discussion that tubercle neurons "encodes odor salience rather than signed odor valence". I'm not sure that I appreciate how the authors results directly disambiguate this, nor am I even clear how one could do in the context of like experiments.

*Reviewer #2 (Recommendations for the authors):*

1) It is unclear how neuronal responses to neutral odors are taken into consideration when calculating valence scores. For example, a neuron with a strong (positive) valence score would be expected to respond to the rewarded odor but not to the neutral odor or odors associated with the air puff. The current analysis appears to ignore neutral odors and should be expanded to include neutral odors.

2) In figure 3, the authors claim that for D1 but not D2 neurons, the valence score increases with training. Valence scores in naive mice are expected to be zero – this should be reflected in the data. To help interpretability, the distributions of valence scores with shuffled cell-odor pairs should be included as reference points.

Furthermore, figure 1 suggests that there is a rage of anticipatory lick numbers on day one, likely reflecting differences in learning during the first session. When analyzing imaging data over time, the authors should compare data from mice with similar behavioral performance.

3) In figure 4, the authors state that the mean valence score of D1 neurons remains above zero for block 2 trials – the mean valence score should be indicated in the figure. Statistical tests for comparing odor valence and identity distributions should be described in more detail.

4) The motivation for the odor-tone association experiments described in figure 5 is unclear. Without characterizing tone-outcome learning in OT neurons in the described experimental setting, the multimodal task is difficult to interpret. Furthermore, the response properties of neurons in the 'full task condition' are hardly analyzed (supplementary figure 4), instead, the analysis focuses on block 2 trials and is difficult to follow.

The main result is that different subpopulations of OT neurons encode odor and tone associations. However, this does not appear to be a question the authors set out to address and may be a trivial result given the well-established differences in neuronal connectivity.

5) It is unclear how odor identity (and sound frequency) is represented in neuronal response patterns in figures 2 and 4. The authors should show neuronal responses grouped by mice / odorant-unconditioned stimulus combinations, in addition to showing neuronal responses grouped by the trial outcomes only.

*Reviewer #3 (Recommendations for the authors):*

p. 2

The term "addictive" is not appropriate to describe intracranial self-administration of cocaine. Use "reinforcing" or "rewarding", instead.

Suppl. Figure 1A

OT D1R neurons do not project to the substantia nigra or ventral tegmental area, or OT D2R neurons do not project to the globus pallidus. It is confusing to mention such projections that are most likely caused by viral diffusions to the nucleus accumbens and dorsal striatum. Clarify the points that the authors are making here.

Figure 3A

Dotted circles used to describe their identifies are not shown on the actual diagram.

p.3

Licking response may not be labeled as "instrumental". Essentially, the study ued a Pavlovian conditioning procedure rather than operant conditioning procedure. Therefore, the conditioned licking response is a Pavlovian conditioned response.

p. 4

For clarity, rewrite the sentence that contains the phrase "non-zero licks were excluded."

Rewrite the sentence "Odors 1 and 5, which... were combined with two sound tones". This sentence does not convey what the authors are trying to describe. The two modalities were combined in the new experiment; odors did not get combined with tones, which has a completely different meaning.

p. 4 and Figure 5A,

Avoid the phrases "odor and sound pairs" and "odor-sound pairs". In the psychology/behavior conditioning literature, they mean that odors are paired with sounds.

Figure 5B

The plot suggests that by day 3, the mice may not have fully discriminated between sounds 1 and 2. The mice displayed anticipatory licks with sound 1 while they did not with odor 1.

Figure 5F-5G and Suppl. Figure 5B-5C

It is more informative to show data separately between O and S for the plots of the OS experiment.

Suppl. Figure 4

Airpuff-paired sound triggered little or no conditioned response from the D1 neurons. What does this mean?

Suppl. Figure 5B-5C

The Y-axis is not labeled.

p. 6

What do water- or airpuff-paired neural signals encode when they are present in the absence of behavior?

The last full paragraph on p. 6

The discussion is confusing. The authors mention that D1R neurons and D2R neurons have differential projections and that they have confirmed the fact in Fg. S1A. However, the authors do not discuss how projection patterns of the D1R and D2R neurons in the dorsal striatum and the nucleus accumbens are relevant to those of the OT. OT D1R and D2R neurons primarily project to the ventral pallidum (Heimer et al. 1987, Zhou et al. 2003); it is not clear how these neurons differentially affect downstream circuits.

Heimer L, Zaborszky L, Zahm DS, Alheid GF. 1987. The ventral striatopallidothalamic projection: I. The striatopallidal link originating in the striatal parts of the olfactory tubercle. J Comp Neurol 255: 571-91

Zhou L, Furuta T, Kaneko T. 2003. Chemical organization of projection neurons in the rat accumbens nucleus and olfactory tubercle. Neuroscience 120: 783-98

The paragraph starting from the bottom of p. 6 and ending at the top of p. and

It is not coherently written. It is hard to follow the arguments.

p. 7

The authors state "D1 OT neurons selectively and bidirectionally encode learned odor valence, unlike D2 neurons". This statement sounds too strong. The sub-heading in the result section on p. 3 characterizes the results more sensibly as follows: "Individual D1 neurons are more likely to encode odor valence and D2 neurons more likely to encode odor identity".

---

## [Author Response]

Essential revisions:1) Authors should provide more details about the specific cell types being monitored in the D1 and A2A-cre mice since cre is present in neurons, not just medium spiny / striatal projection neurons. The authors did not exclude these other cells in their analysis or experimentally so any revision should be clear about inclusion of other types of neurons (e.g., D3).

We acknowledge the potential heterogeneity of the cells we monitored in these mice. Two concerns raised by Reviewer 3 are: (1) cells in the islands of Calleja (IC), which has been shown to express D1 in addition to D3 receptors may be included in the imaging and (2) we may have inadvertently imaged cells in the ventral pallidum, which is located just dorsal to the OT. We address these concerns in detail in the response to Reviewer 3 below (Weakness, point 2). We add some text in the Discussion, acknowledging these points (Discussion third paragraph).

2) The function of the different subpopulations of 'valence coding' D1 neurons should be clarified in both the Results and Discussion. Relatedly, it is unclear how neuronal responses to neutral odors are taken into consideration when calculating valence scores. It is suggested to include the distributions of valence scores with shuffled cell-odor pairs as reference points. Statistical tests for comparing odor valence and identity distributions should be described in more detail.

We now discuss the different subpopulations (Discussion third paragraph), and we explain in detail in the Response to reviewer 1 (first point). We have also considered different metrics for describing valence that would include neutral odors, and discuss why we converged on the particular metric we use. We explain in detail our response below (Reviewer 2, point 1). We have now used shuffled distributions for comparison (Figure 3F, Figure 4FG, Figure 5FG and the associated text in the Results section) – thanks for this important suggestion. Finally, we describe statistical tests in greater detail throughout the manuscript.

3) It is unclear how odor identity (and sound frequency) is represented in neuronal response patterns. The authors should show neuronal responses grouped by mice / odorant-unconditioned stimulus combinations, in addition to showing neuronal responses grouped by the trial outcomes only.

Thank you for this suggestion. To better illustrate the neuronal responses to specific odor-outcome pairings, we have added a supplementary figure with neurons grouped by mouse / odorant-outcome combination (Figure 2 figure supplement 1 and Figure 5 figure supplements 2 and3). The basic patterns of results hold for individual mice.

4) There was concern about a large portion of the brain damaged by the placement of the GRIN lens. The implication of this damage needs to be discussed.

We have added some text discussing this damage (Discussion, eigth paragraph), which is an unfortunate outcome of the experimental procedure. We address the issue in two ways. First, we argue that the principal findings in our work on the OT neuronal activity are unlikely to be related to this damage. Second, we show that the behavior of the mice is not detectably affected by the damage (Figure 1 figure supplement 5). We discuss these issues in detail below in the response to reviewer 3.

5) It is suggested to better link behavior to neural responses- specifically the timing of licking with the GCaMP response.

We appreciate this suggestion and have compared the timing of licking and the neuronal (calcium) responses. Details are below in the response to Reviewer 1 (forth point). Data are now shown in Figure 1 figure supplement 3.

6) Authors should provide a clearer rationale for the tone learning study. The motivation for the odor-tone association experiments is unclear. The main result is that different subpopulations of OT neurons encode odor and tone associations. However, this does not appear to be a question the authors set out to address a priori.

The tone learning experiments were actually an integral part of the study design. We used tones to determine whether adding another stimulus of a different modality will recruit the same cells for valence coding. We have strong evidence from our earlier study (Millman and Murthy, J Neurosci 2020) that adding new odors does indeed recruit same valence cells, so we wanted to see if adding a rewarding cue from a different modality will engage the same cells. Although the use 4 tones (2 for aversion and 2 for reward) might be even cleaner, nevertheless, our current data indicate that adding a new stimulus does NOT automatically recruit the same reward category cells. We present a detailed response below under Reviewer 2, point 4. We explain these points in the manuscript now (Results section, first paragraph when sound experiments are presented, and Discussion seventh paragraph), and also acknowledge the limitations.

Reviewer #1 (Recommendations for the authors):This study is a much needed addition to the field, and the manuscript is excellent and well done. I have a few number of major and a longer list of minor issues I'd appreciate being addressed.Figure 4F2 – the authors package these results that most D1 neurons maintain valence coding in absence of instrumental responding or conditioned outcomes, and while this is true, a good amount do not. It would be beneficial for the authors to elaborate upon what those two different subpopulations of D1 neurons are doing both in Results and Discussion.

This is a question of great interest to us. In response to the Reviewer’s comment, we conjectured that it may be possible that a subpopulation of licking-responsive D1 neurons with high valence scores in Block 1 may have stopped differentiating between rewarded and unrewarded odors in Block 2. In order to test this, we correlated the activity of each neuron on a trial-by-trial basis in the pre-outcome period to the number of anticipatory licks in the pre-outcome period in Block 1. Activity of individual neurons was considered to be significantly correlated to anticipatory licking if the absolute value of Pearson’s correlation coefficient was > 0.5 and the p value was < 0.05 in either odor 4 or odor 5 trial types (16 trials each odor type). We then compared the distribution of the valence scores of Block 1 licking correlated and non-correlated neurons in Blocks 1 and 2 (see Figure 4—figure supplement 2). In contrast to the hypothesis that licking correlated neurons may be less likely to encode valence in Block 2, in the absence of licking, we found that these neurons maintained high valence scores in Block 2. In both groups of neurons, the distribution of valence scores in Block 2 was significantly different from that of the shuffled distribution (Wilcoxon Rank sum test p < 0.01). Rather, the valence scores of the Block 1 licking correlated neurons were significantly higher in Block 2, in the absence of licking, than those of the non-licking correlated neurons in Block 2 (Wilcoxon Rank sum test p < 0.0001). This suggests that there is not a separate population of neurons responding to licking and valence separately, but rather that licking and valence responsivity are intertwined factors, such that even in the absence of licking the same neurons continue to encode odor valence.

We continue to be interested in the factors that lead to the decrease in the valence coding in Block 2 and suggest that the animal’s low motivational state in Block 2 (sated on water, lack of outcomes, and engagement) likely results in lower overall engagement of the OT neuronal circuitry. In addition, in almost all coding frameworks, not every cell has directly interpretable responses and many factors will clearly influence firing. It is also plausible that any cellular plasticity mechanisms involved in the generation of the firing patterns will have stochastic elements such that not all neurons (and connections) will be modified in the desired direction. Another way to interpret the data is that the particular task we have engages only a small part of the overall capacity of the OT neural circuit – so one can expect only a fraction of the neurons to be altered by this task, leaving many as reservoirs for other functions. We continue to be interested in this question, but more directed experimentation may be required to explain the changes in the neuronal activity levels and valence coding between Blocks 1 and 2 and contrast it with typical representational drift. We have added this data in Figure 4 figure supplement 2B, and discuss this issue in the Discussion section, paragraph 3.

The authors do show that some cells display auditory valence coding, yet conclude across the population there is no such activity. It would be helpful for the authors to describe activity of the cells which do in fact show learned responses to the tones.

In the present version of the study, we only used a single aversive tone and a single rewarding tone. While there were clearly strong neuronal responses to these sound tones, we don’t have multiple sound tones for outcomes of the same valence as we did for odors in the original five odor task. Therefore, we cannot be sure that a cell responding to a rewarding tone is simply because of sensory tuning, or because of valence coding. It is certainly possible that if we had used multiple sound tones of the same valence, we would observe valence coding for sounds as we observed for odors in the five odor task. The primary question we asked with the existing experiments was whether neurons responding to odor of the same valence also responded to auditory stimuli of the same valence, and we found that this was not common – hence the perhaps confusing phrasing of “not modality invariant” (which we have replaced with “modality-sensitive”). We note that our earlier study (Millman and Murthy J Neurosci 2020) offered strong evidence that adding new rewarded odors will indeed recruit same valence cells that respond to familiar rewarded odors. Our current data indicate that adding a new stimulus of a different modality does NOT automatically recruit the same reward category cells.

Related to this, in some places the authors describe their results as "not modality invariant". Perhaps this could be rephrased to read, "modality variant" to avoid double negatives. Finally, isn't the finding that there are responses to multisensory stimuli and that these responses are dependent upon learning reason to reconsider titling this paper as originating from the "olfactory striatum"? These results show this is not simply an "olfactory" structure, and as described in Wesson 2020 (Tubular Striatum), the concept of calling the olfactory tubercle the "olfactory striatum" lost ground decades ago. Of course, I'm all for scholarly creativity of the authors and am not asking them to change the title, but more to think about what message this sends to readers.

We have rephrased the relevant text as “modality-sensitive” (in the Abstract and in the main text). We appreciate the point made by this reviewer about the multisensory nature of this structure and the use of the term olfactory striatum. After considering all alternatives, we prefer to use the classical term olfactory tubercle and have titled the paper as “Distinct representation of cue-outcome association by D1 and D2 neurons in the ventral striatum’s olfactory tubercle**”**

Overall, it would be helpful to see the timing of licking, even if on average, to help relate the gcamp response to behavior better. This is important since the authors used a fixed ITI of 20s and normalized each neuron's activity to a 1s pre-trial baseline. But can we be sure there was no anticipatory-related activity during that time? More specifically, it would be nice to show a new figure/inset of Figure 1C, day 1 and/or 4 in raster-format (similar to 4B perhaps) so readers can gain an appreciation for the timing of the licks specifically in this panel, if possible (it's reported in the first results paragraph that mice began licking after odor onset before water delivery, but it is not shown as such).

Thank you for this excellent suggestion. We have plotted the time of the first lick post-odor onset for each of the four training days in D1 and D2 mice, and the distribution of the times of the first lick (we show this in Figure 1 figure supplement 2). We find that the time of the first lick is on average 0.72s ± 0.3s (std dev) after odor onset. Rarely, the mice would lick immediately after the opening of the odor valve, but this was not in response to only the rewarding odors but occurred also in response to other odor types. The timing of the first lick we observe here is later than had been observed in previous studies (Gadziola et al., 2015, Millman and Murthy, 2020) because in those studies a licking response was required to receive the water reward, whereas in our study the delivery of the water was not contingent on the licking response. Hence, the mice tended to start licking closer to the time of the water delivery, which occurred at 1.3s after the odor onset. The late onset of the licking response that we quantified further strengthens the argument that the valence-related activity we observed in the first 0.5s after odor onset is not a result of the licking response but rather in response to the positively reinforced odors instead. As displayed in Figure 2A and B, the activity of OT neurons often peaks by the 0.5s after odor onset well preceding the licking itself. Per the reviewer’s suggestion, we will place dashed lines in the neuronal population activity plots to indicate the average time of licking onset in the rewarded trials.

I'd appreciate the authors elaborating upon the argument from their Discussion that tubercle neurons "encodes odor salience rather than signed odor valence". I'm not sure that I appreciate how the authors results directly disambiguate this, nor am I even clear how one could do in the context of like experiments.

We propose that salience coding neurons would respond similarly to odors of high relevance (e.g. odors predicting strong airpuff or big water reward) whereas valence coding neurons would respond similarly to odors predicting outcomes of the same sign (e.g. two odors predicting water reward), as in the hypothetical responses illustrated in Figure 3A. Given the distinctly different population and individual neuronal activity to odors predicting positive and negative outcomes, we quickly concluded that it was rare for neurons to exhibit response patterns consistent with salience coding without regard to outcome direction. We quantify this in the following portion of the text “the prevalence of neuronal responses consistent with odor salience coding was very low (3 of 529 D1 neurons were activated for high motivational salience odors 1 and 5, but not odors 2-4, as compared to 51 neurons activated for positive valence odors 4 and 5, but not odors 1-3, p < 0.000001 Fisher’s Test for D1 and D2 neurons).”

In response to the Reviewer’s question, and to further elaborate the difference between valence and salience coding, we added a salience score calculated analogously to the valence score as follows:

Salience Score=mean(differences between different salience odors) − mean(differences between similar salience odors)|maximum response −minimum response|

In this calculation, odor pairs 3-5 and odor pairs 3-1 were considered to have different salience (neutral odor v odor with highly salient outcome). Odor pairs 1-5 and 2-4 were considered of more similar salience because they produced large and medium outcomes, respectively. According to this calculation, a neuron with salience-like responses (as shown for example in Figure 3A, third row) should receive a high positive salience score. In contrast, a neuron with very different responses to outcomes of opposite sign (rewarding v aversive) but similar magnitude, should receive a negative salience score inconsistent with salience coding. As expected, we find that there are very few D1 or D2 neurons in either Block 1 or 2 of the task with have positive salience scores. This indicates that the incidence of neurons with responses consistent with salience coding is highly rare in the OT. If anything, a comparison with the shuffle controls indicates that there is greater than chance occurrence of negative scores (as expected if valence coding is strongly prevalent, skewing the salience scores). We present this data in Figure 3 figure supplement 1 and Figure 4 figure supplement 2A now, and mention in the Results.

Reviewer #2 (Recommendations for the authors):1) It is unclear how neuronal responses to neutral odors are taken into consideration when calculating valence scores. For example, a neuron with a strong (positive) valence score would be expected to respond to the rewarded odor but not to the neutral odor or odors associated with the air puff. The current analysis appears to ignore neutral odors and should be expanded to include neutral odors.

Thank you for this clarifying point. We did not sufficiently address the responses of the neurons in neutral odors in the original manuscript and have added results to better demonstrate the neutral odor responses, which are interesting. Neuronal responses to neutral odor trials were not always intermediates between aversive and rewarding odor responses. Instead, neutral odor responses often tracked closely with the aversive odors, and more rarely tracked closely with the rewarded odors or fell between the two (see examples in Figure 3D of our manuscript). In Block 1 of the last day of training, valence responsive neurons (those which clearly differentiated between rewarded and aversive odors) tended to respond to neutral odors similarly as to the aversive odors (panel A in Figure 4—figure supplement 3, left peak in distribution); however, there appeared to be a second smaller group of neurons which responded to neutral odors similarly as they did to the rewarded odors. This was also true of D2 neurons in Block 1 which differentiated between the rewarded and aversive odors (see panel C, Figure 4—figure supplement 3). In Block 2, when the licking and outcomes were omitted, this distribution shifted to the right indicating that neutral odor responses tended to be an intermediate between aversive and rewarding odor responses or track more closely with the rewarded odor responses (see panel B, Figure 4—figure supplement 3). This suggests that the similarity in the responses to the aversive and neutral odors in Block 1 was likely in part due to the fact that the mice were not licking in response to the aversive and neutral odors, but were licking in response to the rewarding odors. The same trend is observed in the principal components of the D1 neuronal activity in Blocks 1 and 2 (Figure 5H in manuscript), where neutral odor responses trend with aversive odor responses in Block 1, but fall intermediate between rewarded and aversive odor responses in Block 2.

In determining the formula for the valence score calculation, we aimed to identify neurons which clearly differentiated between the two aversive and the two rewarding odors.

valence score= mean(differences\ between\ opposite\ valence\ odors)−mean(differences between same valence odors)|maximum response−minimum response

Due to the variability in the neutral odor responses which often tracked closely with either the aversive odors or the rewarded odors in neurons which clearly differentiated between the rewarded and aversive odors, the neutral odor responses were not included in the numerator of the valence score formula. However, while the numerator of the equation does not incorporate neutral odor responses, the denominator would take into account neutral odor responses if they were either the minimum or maximum of those observed. We have added these points in the Methods section when describing valence scores, and present the panels in Figure 4 figure supplement 3.

2) In figure 3, the authors claim that for D1 but not D2 neurons, the valence score increases with training. Valence scores in naive mice are expected to be zero – this should be reflected in the data. To help interpretability, the distributions of valence scores with shuffled cell-odor pairs should be included as reference points.

To provide a point of comparison, we have now incorporated the Reviewer’s great suggestion of the valence scores calculated with shuffled odors in all of the valence score distribution plots (Figure 3, 4, 5). As the identity scores were calculated on the basis of the best odor response as compared to the next best one, the shuffling procedure does not affect the identity scores and was not done for those distributions. These are now shown in Figures 3F, 4F and 5FG, and mentioned in the text.

With regards to the naïve mice – we found that the mice learned the odor association task quickly, such that by the second half of the training session of the first day of training with the odors the mice had learned the associations (Figure 1C and D). Due to the very small number of trials during which the mice were exposed to the odors but had not yet learned the odors associations (< 10 trials per odor), we are unfortunately unable to calculate neuronal valence coding in the naïve state (see individual mice licking rates in Figure 1 figure supplement 3).

Furthermore, figure 1 suggests that there is a rage of anticipatory lick numbers on day one, likely reflecting differences in learning during the first session. When analyzing imaging data over time, the authors should compare data from mice with similar behavioral performance.

The added variability in the day 1 licking rates is in part due to the learning process, but is likely also because in two of the six Drd1-Cre mice we imaged, in the first training day the lick detection did not pick up any licks due to a malfunction (thus those two mice were not included in the day 1 mean). This only occurred on the first day of two D1 mice and not on any other training days or mice. The individual licking rates of the four remaining D1 mice (Figure 1 figure supplement 3) all indicate that the mice began to learn the odor associations within the first day to varying degrees. While we would have liked to have imaging data to analyze at a stage when the mice have not yet learned the odor associations, they learned the task too quickly within < 10 trials of the first session in most cases.

Based on the Reviewer’s suggestion, we plotted the anticipatory licking of each mouse separately across training days (Figure 1 figure supplement 3). We found that by day 2 all mice exhibited clear knowledge of the rewarded odors, and this was maintained through day 4. Note that the mice were not required to lick to receive the water reward, and there were no consequences for licking in response to the aversive and neutral odors. The absolute anticipatory licking rates varied slightly and were modulated by many factors beyond odor-outcome learning including the motivational level (thirst) of the mouse and overall tendency of the mouse to lick. We also confirmed that the mice differentiate between the small water reward predicting odor and the large water reward predicting odor. In day 4 of training, they did an average of 2.and anticipatory licks in response to odor 5 and an average of 2.0 anticipatory licks in response to odor 4 (p < 0.0001 Wilcoxon Rank Sum test, 30 trials each odor in 6 mice). This suggests that the mice learned the gradient of odor outcomes rather than simply categorizing odors into rewarding or aversive.

Due to the quick learning and ceiling effect of the behavior, we only compare neuronal activity in days 1 and 4 in our analysis. The three main neuronal imaging results we report (D1 and D2 neuron type comparison – Figure 2 and 3, Block 1 and sated Block 2 comparison – Figure 4, and sound-odor v odor-odor task comparison – Figure 5) are all from imaging experiments conducted at the end of the training. By day 4, all mice clearly differentiated between the rewarded and unrewarded odors (see day 4 in Figure 1—figure supplement 3) and we concluded that it is reasonable to combine the neuronal activity recorded from the mice. We show the individual mouse licking data in Figure 1 figure supplement 3.

3) In figure 4, the authors state that the mean valence score of D1 neurons remains above zero for block 2 trials – the mean valence score should be indicated in the figure. Statistical tests for comparing odor valence and identity distributions should be described in more detail.

Thank you for this suggestion. The mean valence score for D1 neurons in Block 1 is 0.39, the mean valence score for D1 neurons in Block 2 is 0.18 and there is a bimodal distribution of valence scores in Block 2 with one of the modes centered around zero and one of the modes centered around 0.4 similar to the mean in Block 1. The non-parametric two-tailed Wilcoxon rank sum test was used in all statistical comparisons of valence and identity score distributions. The Wilcoxon signed rank test indicates that the distribution of the overall valence scores of D1 neurons in Block 2 is significantly different from zero p = 2.8151e-13. Per the Reviewer’s suggestion, we also added the shuffled valence score distribution for the same neurons and find that the Block 2 D1 neuron valence score distribution is significantly different from the shuffled distribution (p = 2.1025e-12, Wilcoxon Rank Sum test). We have added this information in the main text in the Results section (rather than in the figure legends, to avoid clutter).

4) The motivation for the odor-tone association experiments described in figure 5 is unclear. Without characterizing tone-outcome learning in OT neurons in the described experimental setting, the multimodal task is difficult to interpret. Furthermore, the response properties of neurons in the 'full task condition' are hardly analyzed (supplementary figure 4), instead, the analysis focuses on block 2 trials and is difficult to follow.The main result is that different subpopulations of OT neurons encode odor and tone associations. However, this does not appear to be a question the authors set out to address and may be a trivial result given the well-established differences in neuronal connectivity.

Thank you for raising this point which was not previously explained well. We had three motivations for conducting the sound-odor association experiment. First, it has previously been demonstrated that neurons in the tubercle can become activated in response to auditory tones (Varga and Wesson, 2013, Wesson and Wilson, 2010), so we wondered whether the valence related activation we observed in response to odors extended to auditory stimuli of the same valence as well. Second, as we learned that this was not the case, we found that the sound trials could serve as an additional control for checking whether the valence-related neuronal activity we observed was due to the instrumental licking response. The mice licked at similar rates in response to the rewarding odor and sound stimuli, and yet we did not observe the same similarity in neuronal responses to these odor and sound stimuli as we did to odors of the same valence in the five-odor task. This strengthens the argument that the odor valence-related activity we record in the original association task is not purely a result of the shared licking response to the two rewarded odors. Third, in the long term, we are interested in investigating the question of whether odor associations have unique qualities such as being more long-lived or more likely to elicit strong emotional responses due to the direct convergence of olfactory bulb input and dopaminergic input in limbic brain regions such as the olfactory tubercle. For this reason, we found it useful to use sound associations and related neuronal responses to gather evidence to begin to address this question. We clarify this in the manuscript in the Results section, when we first describe the odor-sound experiments and Figure 4.

The reviewer has correctly pointed out that in the main text we selected to focus on the neuronal imaging results in the final blocked version of the sound-odor association task. This was done primarily for the sake of brevity, to focus on the most direct comparison of the neuronal responses to the stimulus-only condition to the sounds and odors. In doing so, we could directly compare valence coding in the stimulus only condition in the odor-odor task and in the odor-sound task and eliminate the effects of the licking responses and the water/airpuff outcomes which were shared in the two tasks. We do display the distribution of the valence scores in the full-task condition in the odor-odor and odor-sound tasks in Figure 5F, the Block 1 and Block 2 neuronal activity principal components in Figure 5I, and the full-task neuronal activity in Figure 5 figure supplement 1. We also show the cosine similarity for the O-O and O-S task in Blocks 1 and 2 in Figure 5J.

5) It is unclear how odor identity (and sound frequency) is represented in neuronal response patterns in figures 2 and 4. The authors should show neuronal responses grouped by mice / odorant-unconditioned stimulus combinations, in addition to showing neuronal responses grouped by the trial outcomes only.

Thank you for this suggestion. To better illustrate the neuronal responses to specific odor-outcome pairings, we have added a supplementary figure with neurons grouped by mouse / odorant-outcome combination. The basic patterns of results hold for individual mice. These are shown in Figure 2 figure supplement 1 and Figure 5 figure supplement 1.

Reviewer #3 (Recommendations for the authors):p. 2The term "addictive" is not appropriate to describe intracranial self-administration of cocaine. Use "reinforcing" or "rewarding", instead.

We have changed in to “reinforcing”.

Suppl. Figure 1AOT D1R neurons do not project to the substantia nigra or ventral tegmental area, or OT D2R neurons do not project to the globus pallidus. It is confusing to mention such projections that are most likely caused by viral diffusions to the nucleus accumbens and dorsal striatum. Clarify the points that the authors are making here.

We apologize for the confusion. The reviewer is correct in that the OT D1 and D2 neurons do not project to SNc/VTA and GP. This figure was included to validate the mouse models used – that the D1 and D2 mouse lines have differential projections, and the virus spillover in regions near the OT (for example, other ventral striatal regions) will allow us to trace projections to the SNc and VTA. However, this is needlessly confusing, and these mouse lines have been extensively validated previously, therefore we have removed this figure panel.

Figure 3ADotted circles used to describe their identifies are not shown on the actual diagram.

Apologies – we have added these now.

p.3Licking response may not be labeled as "instrumental". Essentially, the study ued a Pavlovian conditioning procedure rather than operant conditioning procedure. Therefore, the conditioned licking response is a Pavlovian conditioned response.

We agree with this assessment – we have changed instances of the term “instrumental” to “licking” in the text.

p. 4For clarity, rewrite the sentence that contains the phrase "non-zero licks were excluded."

OK. We have changed this to “any trials in which the mice licked were excluded”.

Rewrite the sentence "Odors 1 and 5, which... were combined with two sound tones". This sentence does not convey what the authors are trying to describe. The two modalities were combined in the new experiment; odors did not get combined with tones, which has a completely different meaning.

OK, we have changed this to: A new task was introduced in which two odor-outcome associations and two sound-outcome associations were used. The odors 1 and 5 were paired with strong airpuff and large water reward as previously, and two sound tones were used (5kHz and 12kHz) also paired with strong airpuff and large water reward.

p. 4 and Figure 5A,Avoid the phrases "odor and sound pairs" and "odor-sound pairs". In the psychology/behavior conditioning literature, they mean that odors are paired with sounds.

This is a fair point. We have changed the text to remove pairs, and just refer to them as “odor and sound”.

Figure 5BThe plot suggests that by day 3, the mice may not have fully discriminated between sounds 1 and 2. The mice displayed anticipatory licks with sound 1 while they did not with odor 1.

We agree with the Reviewer that there is more anticipatory licking in response to the aversive sound 1 than to the aversive odor 1. However, the mice tend to lick marginally more both in response to the rewarded and aversive sounds – possibly due to the startling nature of the sound onset. In day 3, the mice clearly and strongly discriminated between sounds 1 and 2. They perform an average of 3.and anticipatory licks in response to sound 2 and 1.1 licks in response to sound 1 (p = 8.0914e-28, Wilcoxon rank sum test). The difference between the number of licks between sound 1 and sound 2 is ~2.6 licks. In comparison, the mean number of anticipatory licks for small rewarded odor 4 in the odor-odor task was 2.2 and the number of licks for aversive odor 1 was 0.08, for a smaller difference of 2.1 licks between the two stimuli. Therefore, we are sure that mice are indeed discriminating quite well between sounds 1 and 2. We present these numbers when describing the sound-odor experiments in the Results section.

Figure 5F-5G and Suppl. Figure 5B-5CIt is more informative to show data separately between O and S for the plots of the OS experiment.

We are not sure we understand the reviewer’s point. In the Odor-Sound experiment, there is no “O” valence score or “S” valence score, since there is only one odor and sound for each valence. The valence scores can only be calculated by comparing odors and sounds of the same vs opposite valence.

Suppl. Figure 4Airpuff-paired sound triggered little or no conditioned response from the D1 neurons. What does this mean?

We thank the Reviewer for this interesting observation. While we can only conjecture, we believe the reason for this may be that the activity in response to the sound stimuli in the “full task” condition as shown in Suppl. Figure 4 may be dominated by the response of the mice to the stimuli and the outcomes rather than the sound stimuli themselves. In the case of the airpuff-paired sound, the mice perform little anticipatory licking and likely the most salient event occurs at the onset of the airpuff. This is also supported by the fact that the airpuff-onset related neuronal activity is similar in the aversive odor and aversive sound trials. In the case of the water-paired sound, the mice begin anticipatory licking after the onset of the sound and continue to do so when the water outcome is delivered. However, we also note the presence of a significant number of sound 2 responsive neurons in Block 2 of the odor-sound task when there was no licking, indicating that the neuronal activity in response to the rewarded sound is not only due to licking. Indeed, there were very few (2.and%) of D1 neurons which were preferentially activated to aversive sound stimulus in Block 2. It remains to be explored whether D1 OT neurons are more likely to respond to reinforced sounds than aversive sounds, but our data indicates that this may be the case. We really don’t have anything informative to add in the manuscript, so have not done so.

Suppl. Figure 5B-5CThe Y-axis is not labeled.

Sorry for the oversight, and thanks for noting this. Corrected.

p. 6What do water- or airpuff-paired neural signals encode when they are present in the absence of behavior?

We are not sure what exactly the reviewer is asking, but here is an interpretation. Sensory signals are presumably converted/processed to allow motor outputs. In the earliest stages, the signals are likely to be largely sensory, and the latest stages signals will relate to motor acts (one has to be careful, of course, since plenty of recent evidence suggests that sensory areas have strong motor signals as well). In the intermediate regions, signals are likely to be mixed. We suggest that signals in the OT are strongly sensory-related, but modified substantially to allow for categorical predictions. Therefore, while behavior can affect these signals, there will still be significant sensory component even in the absence of behavior (after all the OT gets strong inputs from the OB and piriform cortex).

The last full paragraph on p. 6The discussion is confusing. The authors mention that D1R neurons and D2R neurons have differential projections and that they have confirmed the fact in Fg. S1A. However, the authors do not discuss how projection patterns of the D1R and D2R neurons in the dorsal striatum and the nucleus accumbens are relevant to those of the OT. OT D1R and D2R neurons primarily project to the ventral pallidum (Heimer et al. 1987, Zhou et al. 2003); it is not clear how these neurons differentially affect downstream circuits.Heimer L, Zaborszky L, Zahm DS, Alheid GF. 1987. The ventral striatopallidothalamic projection: I. The striatopallidal link originating in the striatal parts of the olfactory tubercle. J Comp Neurol 255: 571-91Zhou L, Furuta T, Kaneko T. 2003. Chemical organization of projection neurons in the rat accumbens nucleus and olfactory tubercle. Neuroscience 120: 783-98

Once again, we apologize for the confusing presentation. As noted in the response above to the reviewer’s earlier comment, we do not mean to imply that OT neurons project to VTA/SNc etc. We used Supp Figure 1A to validate the mouse lines used, and we realize that this is confusing/misleading. We have removed Supp Figure 1A and its reference – instead we use the references suggested by the reviewer to make our differential projections point in the Discussion (paragraph 5).

The paragraph starting from the bottom of p. 6 and ending at the top of p. andIt is not coherently written. It is hard to follow the arguments.

We have tried to clarify this part, in Discussion (paragraph 6) now.

p. andThe authors state "D1 OT neurons selectively and bidirectionally encode learned odor valence, unlike D2 neurons". This statement sounds too strong. The sub-heading in the result section on p. 3 characterizes the results more sensibly as follows: "Individual D1 neurons are more likely to encode odor valence and D2 neurons more likely to encode odor identity".

We have modified this sentence to “D1 OT neurons are more likely to encode learned odor valence than D2 neurons, and conversely less likely to encode odor identity”.

References:

Barik, S. and de Beaurepaire, R. (1998). Hypothermic effects of dopamine D3 receptor agonists in the island of Calleja Magna. Potentiation by D1 activation. *Pharmacol Biochem Behav* 60, 313-9.

Cowley, B. R., Kaufman, M. T., Butler, Z. S., Churchland, M. M., Ryu, S. I., Shenoy, K. V. and Yu, B. M. (2013). DataHigh: graphical user interface for visualizing and interacting with high-dimensional neural activity. *J Neural Eng* 10, 066012.

Mansour, A., Meador-Woodruff, J. H., Bunzow, J. R., Civelli, O., Akil, H. and Watson, S. J. (1990). Localization of dopamine D2 receptor mRNA and D1 and D2 receptor binding in the rat brain and pituitary: an in situ hybridization-receptor autoradiographic analysis. *J Neurosci* 10, 2587-600.

Mengod, G., Villaro, M. T., Landwehrmeyer, G. B., Martinez-Mir, M. I., Niznik, H. B., Sunahara, R. K., Seeman, P., O'Dowd, B. F., Probst, A. and Palacios, J. M. (1992). Visualization of dopamine D1, D2 and D3 receptor mRNAs in human and rat brain. *Neurochem Int* 20 Suppl, 33S-43S.